 TOOLS AND RESOURCES

# A precisely adjustable, variation-suppressed eukaryotic transcriptional controller to enable genetic discovery

**Asli Azizoglu[1]\*, Roger Brent[2]†\*, Fabian Rudolf[1]†\***

[1]Computational Systems Biology and Swiss Institute of Bioinformatics, ETH Zurich, Basel, Switzerland; [2]Division of Basic Sciences, Fred Hutchinson Cancer Research Center, Seattle, United States

**Abstract** Conditional expression of genes and observation of phenotype remain central to biological discovery. Current methods enable either on/off or imprecisely controlled graded gene expression. We developed a 'well-tempered' controller, $WTC_{846}$, for precisely adjustable, graded, growth condition independent expression of genes in *Saccharomyces cerevisiae*. Controlled genes are expressed from a strong semisynthetic promoter repressed by the prokaryotic TetR, which also represses its own synthesis; with basal expression abolished by a second, 'zeroing' repressor. The autorepression loop lowers cell-to-cell variation while enabling precise adjustment of protein expression by a chemical inducer. $WTC_{846}$ allelic strains in which the controller replaced the native promoters recapitulated known null phenotypes (*CDC42, TPI1*), exhibited novel overexpression phenotypes (*IPL1*), showed protein dosage-dependent growth rates and morphological phenotypes (*CDC28, TOR2, PMA1* and the hitherto uncharacterized *PBR1*), and enabled cell cycle synchronization (*CDC20*). $WTC_{846}$ defines an 'expression clamp' allowing protein dosage to be adjusted by the experimenter across the range of cellular protein abundances, with limited variation around the setpoint.

**\*For correspondence:**
asli.azizoglu@bsse.ethz.ch (AA);
rbrent@fhcrc.org (RB);
fabian.rudolf@bsse.ethz.ch (FR)

†These authors contributed equally to this work

**Competing interests:** The authors declare that no competing interests exist.

## Introduction

Since the spectacular demonstration of suppression of nonsense mutations and its application to T4 development (*Epstein et al., 1963*), means to express genes conditionally to permit observation of the phenotype have remained central to biological experimentation and discovery. During the 20th century, workhorse methods to ensure the presence or absence of gene products have included use of temperature-sensitive (*ts*) and cold-sensitive (*cs*) mutations within genes, for example to give insight into ordinality of cell biological events (*Hereford and Hartwell, 1974*). After the advent of recombinant DNA methods, conditional expression of genes into proteins, for example by derepression of *lac* promoter derivatives (*Goeddel et al., 1979*), also found application in biotechnology for production of therapeutics and industrial products (*Sochor et al., 2015*). In 2021, contemporaneous approaches to conditional expression in wide use include construction of transgenes activated by chimeric activators controlled by promoters whose expression is temporally and spatially restricted to different cell lineages (*Brand and Perrimon, 1993*), hundreds of approaches based on production of DNA rearrangements by phage-derived site specific recombination (*Sauer, 1987*), and triggered induction of engineered genes by chimeric transcription regulators with DNA-binding moieties based on derivatives of TetR from Tn10 (*Gossen et al., 1995*; *Garí et al., 1997*). Most of these approaches are all-or-none, in the sense that they are not intended to bring about expression of intermediate levels of protein; and the observations they enable are often qualitative.

But it has long been recognized that adjustment of protein dosage can provide additional insight into function that cannot be gained from all-or-none expression. For example, controlled expression

of the bacteriophage λ cI and cro gene products was key to understanding how changes in the level of those proteins regulated the phage's decision to undergo lytic or lysogenic growth (*Maurer et al., 1980*; *Meyer et al., 1980*; *Meyer and Ptashne, 1980*). In *S. cerevisiae*, contemporaneous means to tune dosage include metabolite induced promoters, such as $P_{GAL1}$, $P_{MET3}$, $P_{CUP1}$ (*Maya et al., 2008*), in which expression is controlled by growth media composition, and small molecule induced systems, such as the β-estradiol-induced LexA-hER-B112 system (*Ottoz et al., 2014*). Many of these depend on fusions between eukaryotic and viral activator domains and prokaryotic proteins (*Garí et al., 1997*; *Ottoz et al., 2014*; *McIsaac et al., 2013*; *McIsaac et al., 2014*) that bind sites on engineered promoters (*Brent and Ptashne, 1985*). These methods suffer from a number of drawbacks, including basal expression when not induced (*Bellí et al., 1998*; *Garí et al., 1997*; *Ottoz et al., 2014*), deleterious effects on cell growth due to sequestration of cellular components by the activation domain (*Gill and Ptashne, 1988*) induction of genes in addition to the controlled gene (*McIsaac et al., 2013*), and high cell-to-cell variation in expression of the controlled genes (*Meurer et al., 2017*; *Elison et al., 2017*; *Ottoz et al., 2014*).

These inducible systems rely on 'activation by recruitment' (*Ptashne and Gann, 1997*); the activator binds a site on DNA upstream of a yeast gene and recruits general transcription factors and regulators of the Pre-Initiation Complex (PIC). These assemble downstream at the 'core promoter' and recruit RNA polymerase II to induce transcription (*Hahn and Young, 2011*). An alternative to inducible activation would be to engineer reversible repression of yeast transcription by prokaryotic repressors (*Brent and Ptashne, 1984*; *Hu and Davidson, 1987*; *Brown et al., 1987*). For TATA-containing promoters, binding of prokaryotic proteins such as LexA and the lac repressor near the TATA sequence can repress transcription (*Brent and Ptashne, 1984*; *Murphy et al., 2007*; *Wedler and Wambutt, 1995*), presumably by interference with the formation of the PIC, transcription initiation, or early elongation. It has long been recognized (*Brent, 1985*) that prokaryotic repressors likely work through different mechanisms than mechanisms used by repressors native to eukaryotes (*Wang et al., 2011*; *Gaston and Jayaraman, 2003*).

We envisioned that an ideal conditional expression system to support genetic and quantitative experimentation would: (1) function in all growth media, (2) be inducible by an exogenous small molecule with minimal other effects on the cell, (3) manifest no basal expression of the controlled gene in absence of inducer, allowing generation of null phenotypes, (4) enable a very large range of precisely adjustable expression, and (5) drive very high maximum expression, allowing generation of overexpression phenotypes. Moreover, since differences in global ability to express genes into proteins (*Colman-Lerner et al., 2005*) lead to differences in allelic penetrance and expressivity (*Burnaevskiy et al., 2019*), the ideal controller should (6) exhibit low cell-to-cell variation at any set output, facilitating detection of phenotypes that depend on thresholds of protein dosage, and other inferences of single-cell behaviors from population responses.

Here, we describe the development of a prokaryotic repressor-based transcriptional controller of gene expression, Well-tempered Controller$_{846}$ (WTC$_{846}$), that fulfils the criteria outlined above. This development had three main stages. We first engineered a powerful eukaryotic promoter that is repressed by the prokaryotic repressor TetR and induced by the chemical tetracycline and its analogue Anhydrotetracycline (aTc), to use as the promoter of the controlled gene. Next, we used instances of this promoter to construct a configuration of genetic elements that show low cell-to-cell variation in expression of the controlled gene, by creating an autorepression loop in which TetR repressed its own synthesis. Third and last, we abolished basal expression of the controlled gene in the absence of the inducer, by engineering a weakly expressed 'zeroing' repressor, a chimera between TetR and an active yeast repressor Tup1. With WTC$_{846}$, adjusting the extracellular concentration of aTc can precisely set the expression level of the controlled gene in different growth media, over time and over cell cycle stage. The gene is then 'expression clamped' with low cell-to-cell variation at a certain protein dosage, which can range from undetectable to greater abundance than wild type. We showed that strains carrying WTC$_{846}$ allelic forms of essential genes recapitulated known knockout phenotypes, and one demonstrated a novel overexpression phenotype. We constructed strains bearing WTC$_{846}$ alleles of genes involved in size control, growth rate, and cell cycle state and showed that these allowed precise experimental control of these fundamental aspects of cell physiology. We expect that WTC$_{846}$ alleles will find use in biological engineering and in discovery research, in assessment of phenotypes now incompletely penetrant due to cell-to-cell variation of

the causative gene, in hypothesis directed cell biological research, and in genome-wide studies such as gene-by-gene epistasis screens.

## Results

### Construction of a repressible $P_{TDH3}$ promoter

Our goal was to engineer efficient repression of eukaryotic transcription by a bacterial repressor. We started with a strong (*Ho et al., 2018*), well-characterized, constitutive, and endogenous yeast promoter. This promoter, $P_{TDH3}$, has three key Transcription Factor (TF) binding sites, one for Rap1 and two for Gcr1 (*Yagi et al., 1994*; *Kuroda et al., 1994*) in its Upstream Activating Region (UAS), and a

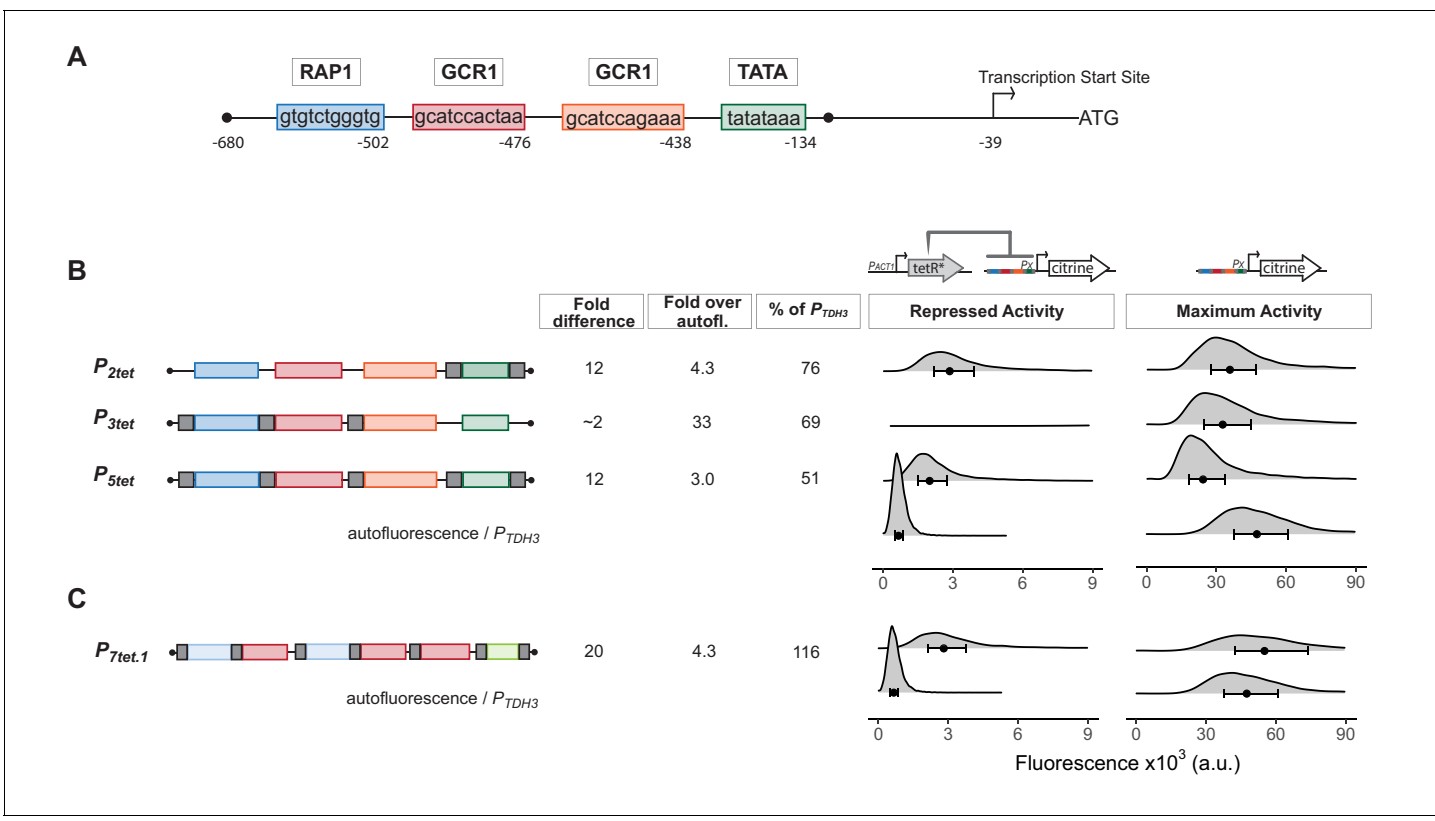

**Figure 1.** Repression of engineered $P_{TDH3}$ derivatives by TetR. (**A**) Structure of the starting promoter, $P_{TDH3}$. Diagram shows the nucleotide positions of the binding sites for the endogenous transcription factors Rap1 and Gcr1, the TATA-sequence, and the transcription start site relative to the start codon of the *TDH3* gene. (**B**) Repression and maximum activity of engineered $P_{TDH3}$ derivatives. Diagrams above the plots display the genetic elements of strains used in B and C. Left diagram depicts strains used to test repressed activity, right diagram maximum activity. Px denotes any TetR repressible promoter. The * in TetR indicates a SV40 Nuclear Localization Sequence. In all strains, the $P_{TDH3}$ derivative promoters diagrammed on the left directed the synthesis of Citrine integrated into the *LEU2* locus. Grey boxes inside the diagrams denote $tetO_1$ TetR-binding sites. For measurement of repressed activity, an additional $P_{ACT1}$-directed TetR was integrated into the *HIS3* locus. Citrine fluorescent signal was detected by flow cytometry. Fold difference refers to the median of the maximum activity divided by the median of the repressed activity signal. Fold over autofluorescence refers to median repressed activity signal divided by the median autofluorescent background signal. Maximum promoter activity is quantified as percentage of $P_{TDH3}$ signal using the medians. x axis shows intensity of fluorescence signal. Plots are density distributions of the whole population, such that the area under the curve equals 1 and the y axis indicates the proportion of cells at each fluorescence value. The circles inside each density plot show the median, and the upper and lower bounds of the bar correspond to the first and third quartiles of the distribution. Repressed activity of $P_{3tet}$ is above the x axis depicted in this figure, but can be seen in *Figure 1—figure supplement 1*. (**C**) Repression and maximum activity of the optimized $P_{7tet.1}$. Diagrams and plots as in (**B**). $P_{7tet.1}$ contained additional binding sites for Rap1 and Gcr1 selected for higher activity, as well as an alternative TATA sequence as described in the Supplementary Information. It shows the highest fold difference, maximum activity comparable to $P_{TDH3}$, and low repressed activity. The online version of this article includes the following source data and figure supplement(s) for figure 1:

**Source data 1.** Numerical data for *Figure 1*.

**Figure supplement 1.** A promoter with three $tetO_1$ sequences in the UAS of $P_{TDH3}$ is only minimally repressed by TetR.

TATA sequence at which PIC assembles on the core promoter (*Figure 1A*). Based on earlier work, we knew that binding of prokaryotic repressors to sites flanking the TATA sequence of $P_{TDH3}$ repressed activity of this promoter (*Wedler and Wambutt, 1995*), presumably by interfering with PIC formation, transcription initiation, or early elongation. We therefore placed well characterized, 15 bp long TetR-binding sites ($tetO_1$) (*Bertram and Hillen, 2008*) immediately upstream and downstream of the $P_{TDH3}$ TATA sequence to create $P_{2tet}$. To determine whether repressor binding could also block function in the UAS, we placed a single $tetO_1$ directly upstream of each Rap1 and Gcr1-binding site to create $P_{3tet}$. We also combined the operators in these constructs to generate $P_{5tet}$ (*Figure 1B*). We integrated a single copy (*Gnügge et al., 2016*) of constructs bearing these promoters directing the synthesis of the fluorescent protein Citrine into the *LEU2* locus (*Griesbeck et al., 2001*).

We compared the Citrine fluorescence signal (measured by flow cytometry at wavelengths 515–545 nm) from these promoters to quantify their activity. We compared the strains Y2551[$P_{2tet}$], Y2564[$P_{3tet}$], and Y2566[$P_{5tet}$] with an otherwise-isogenic strain in which Citrine was expressed from native $P_{TDH3}$ (Y2683). This fluorescence signal measures Citrine expression, but also includes auto-fluorescent background from the yeast cells. We quantified this background by using the otherwise-isogenic parent strain Y70. Measured in this way, $P_{2tet}$ had 76%, $P_{3tet}$ 69%, and $P_{5tet}$ 51% of $P_{TDH3}$ activity (*Figure 1B*). To assess repressibility of these promoters, we compared Citrine expression in these strains with expression in otherwise-isogenic strains in which a genomically integrated $P_{ACT1}$ promoter drove constitutive expression of TetR (Y2562, Y2573, Y2577). By this measure, TetR repressed $P_{2tet}$ by a factor of 12, $P_{3tet}$ by a factor of 1.5, and $P_{5tet}$ by a factor of 12 (*Figure 1B* and *Figure 1—figure supplement 1*). Absolute repressed signal from these promoters was 4.3, 33, and 3 times the autofluorescence background. Because our aim was to create a promoter with no expression when repressed, we viewed even small reductions in repressed expression as useful and therefore decided to use $P_{5tet}$ as a basis for further constructions.

Insertion of $tetO_1$ sites in $P_{TDH3}$ to create $P_{5tet}$ had reduced promoter maximum activity considerably. In order to regain the lost activity, we tested numerous constructs to find optimal placement for the $tetO_1$ sites, optimized Rap1, Gcr1, and TATA sequences, and increased the number of Rap1 and Gcr1 sequences (see Appendix 1 and *Appendix 1- Figure 1*). This work resulted in $P_{7tet.1}$, which carried two Rap1 and three Gcr1 sites, sequence optimized to generate higher promoter activity, and an alternative TATA sequence to that of $P_{TDH3}$. By the assays described above, the new promoter $P_{7tet.1}$ (Y2661) showed comparable maximum expression to $P_{TDH3}$, 20-fold repression of Citrine signal, and absolute repressed activity (Y2663) of 4.3-fold over background (*Figure 1C*). We chose $P_{7tet.1}$ as the promoter to develop our controller with.

## Complex autorepressing (cAR) controller architecture expands the input dynamic range and reduces cell-to-cell variation

We set out to optimize control of genes by $P_{7tet.1}$. To do so, we tested the ability of different constructions that directed the synthesis of TetR to regulate $P_{7tet.1}$-*citrine* directed fluorescence signal. *Figure 2A* shows the three different architectures. In Simple Repression (SR), the $P_{7tet.1}$ controlled gene was repressed by TetR expressed from a constitutive promoter. In Autorepression (AR), the $P_{7tet.1}$ controlled gene was repressed by TetR expressed from a second instance of $P_{7tet.1}$, therefore creating a negative feedback loop. In Complex Autorepression (cAR), a second TetR gene expressed from a constitutive promoter was added to the AR architecture.

We compared the input-output relationship (i.e. dose response) for the three architectures. To do so, we constructed otherwise-isogenic strains with these architectures in which $P_{7tet.1}$ directed Citrine expression (Y2663, Y2674, and Y2741). We used flow cytometry to quantify Citrine fluorescence signal from all strains 7 hr after addition of different concentrations of aTc and fitted a log logistic model to the median fluorescence (see Materials and methods) (*Figure 2B&C*).

Compared to the SR architecture, the AR architecture showed a more gradual dose response curve and a larger input dynamic range (the range of input doses for which the slope of the dose response curve was non-zero), from 3 to 400 ng/mL vs. 5–80 ng/mL aTc. This same flattening of the response curve and increased input dynamic range in autorepressing, synthetic TetR based eukaryotic systems has been described (*Nevozhay et al., 2009*), and we believe it operates in evolved prokaryotic systems including Tn10 and the *E. coli* SOS regulon, in which the TetR and LexA repressors repress their own synthesis (see Discussion). A broader input dynamic range allows more precise

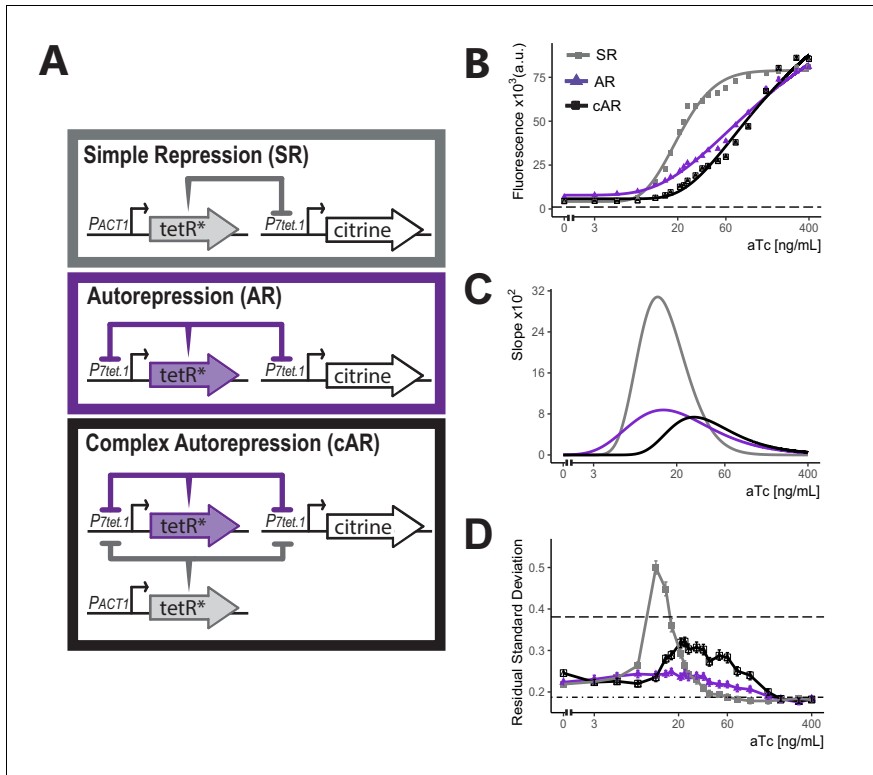

**Figure 2.** Comparison of the three controller architectures. (A) Genetic elements of the different controller architectures used in these experiments. The * next to TetR indicates SV40 Nuclear Localization Sequence and flat headed arrows indicate repression. In all cases, $P_{7tet.1}$ drives Citrine expression integrated at the *LEU2* locus. In SR, the repressor of $P_{7tet.1}$, tetR, is integrated at the *HIS3* locus and is constitutively expressed. In AR, tetR is again integrated at the *HIS3* locus, but is now expressed by $P_{7tet.1}$. cAR has the same constructs as AR and an additional, constitutively expressed zeroing repressor integrated at the *URA3* locus. (B) aTc dose response curves of Citrine expression for the three different architectures. Citrine fluorescence from strains bearing these architectures was measured at steady state using flow cytometry after 7 hr of induction with different concentrations of aTc. Symbols indicate the median fluorescence at each dose. Lines are fitted using a five-parameter log logistic function as explained in Materials and methods. Dashed line indicates autofluorescence signal measured from the parental strain without Citrine. (C) Slopes of the dose response curves in (B). The x axis range with non-zero slopes defines the useful input dynamic range. (D) Cell-to-cell variation of expression by these three architectures. We calculated single-reporter cell-to-cell variation (VIV) as described. Higher Residual Standard Deviation (RSD) values (y axis) correspond to greater VIV. Dot-dash line indicates the VIV of the strain where Citrine is constitutively expressed from $P_{TDH3}$ and dashed line indicates VIV of autofluorescence in the parent strain without Citrine. Error bars indicate 95% confidence interval calculated using bootstrapping (n=1000) as described in Materials and methods. The online version of this article includes the following source data and figure supplement(s) for figure 2:

**Source data 1.** Numerical data for *Figure 2* panels B and D.
**Figure supplement 1.** Variation in expression for the SR, AR, and cAR architectures.
**Figure supplement 1—source data 1.** Numerical data for *Figure 2—figure supplement 1*.
**Figure supplement 2.** Single-reporter VIV measure of CCV in expression.
**Figure supplement 3.** Autorepression loop reduces cell-to-cell variability at intermediate concentrations of aTc.
**Figure supplement 4.** Single-reporter VIV measure of variation in expression from native yeast promoters.
**Figure supplement 4—source data 1.** Numerical data for *Figure 2—figure supplement 4*.

adjustment of protein levels, since small differences in inducer concentration (due for example to experimental errors, or differences in aTc uptake among cells) have smaller effects.

In these experiments, we also measured cell-to-cell variation (CCV) in the expression of the controlled gene. Many existing inducible gene expression systems show considerable variation in expression of the controlled gene, making it difficult to achieve homogenous phenotypes at the population level (*Meurer et al., 2017*; *Elison et al., 2017*; *Ottoz et al., 2014*). In *S. cerevisiae* and

*C. elegans*, comparison of signals from strains with different constellations of reporter genes allows quantification of different sources of variation in protein dosage (*Colman-Lerner et al., 2005*; *Pesce et al., 2018*; *Mendenhall et al., 2015*). Here, we quantified overall variation in protein dosage by measuring the Coefficient of Variation (CoV) in fluorescent output from a single reporter (*Figure 2—figure supplement 1*), and we developed a second measure called Volume Independent Variation (VIV) (explained in Appendix 2) that normalized variation in dosage with respect to a key confounding variable, cell volume, to correct for its effect on protein concentration. In VIV, we estimated cell volume by a vector of forward and side scatter signals, and calculated the remaining (Residual) Standard Deviation of the single reporter output after normalization with this estimated volume (*Figure 2D* and *Figure 2—figure supplement 2*). By both measures, strains carrying the SR architecture showed high variation throughout the input dynamic range, with a peak around the mid-point (12 ng/mL aTc). Strains bearing the AR architecture showed low overall CCV, and no peak at intermediate aTc concentrations. This diminution of CCV in synthetic, autorepressing TetR based eukaryotic systems has previously been described (*Becskei and Serrano, 2000*; *Nevozhay et al., 2009*). In the SR architecture, variations in the amount of TetR in different cells cannot be buffered. In the AR architecture, such variations in repressor concentration are corrected for (see Discussion) and variation in expression of the controlled gene is at or around the same level as seen for constitutive expression driven by a number of native promoters (see *Figure 2—figure supplement 4* for variation of commonly used promoters). This reduced cell-to-cell variation is useful for inferring single cell behaviors by observing population level responses (see Discussion).

Compared with cells bearing the SR architecture, otherwise-isogenic cells bearing the AR architecture showed increased basal expression (6.3 vs. 4.1-fold over autofluorescence background). The increased basal expression was a consequence of the fact that in the AR architecture $P_{7tet.1}$ directs the synthesis of both the controlled Citrine gene and of TetR itself, so that, in uninduced cells, the steady state abundance of TetR was lower than in cells in which synthesis of TetR was driven by $P_{ACT1}$. More important, in the AR architecture, the fact that some amount of TetR expressed from $P_{7tet.1}$ was needed to repress its own synthesis meant that it would not be possible to abolish $P_{7tet.1}$-driven expression of the controlled gene completely. Since ability to abolish basal expression of the controlled gene was an important design goal, we constructed strains with a third architecture, cAR, in which a different constitutive promoter drove expression of a second TetR gene in order to drive basal expression lower. Compared to otherwise-isogenic AR strains, strains expressing Citrine controlled by the cAR architecture showed reduced basal expression (4.1-fold over autofluorescence), and, compared to the otherwise isogenic SR strain, showed reduced CCV and a more gradual dose response (*Figure 2C&D* and *Figure 2—figure supplement 3*). We therefore picked this cAR architecture for our controller.

## Hybrid repressor abolishes basal expression of $P_{7tet.1}$

To further decrease basal expression in the cAR architecture, we set out to create a more effective TetR derivative. Initially, we followed an approach that increased the size and nuclear concentration of TetR by fusing it to other inert bacterial proteins and nuclear localization sequences, but this approach was not enough to abolish all basal expression (see Appendix 3).

$P_{3tet}$ bears $tetO_1$ sites only in its UAS. The fact that $P_{3tet}$ SR strains only showed weak repression (1.5-fold) suggested that TetR, and other inert derivatives described in the Appendix 3, exerted their effects on $P_{7tet.1}$ mostly by their action at the $tetO_1$ sites flanking the TATA sequence. We thus hypothesized that TetR derivatives that carried native, active yeast repressors might more effectively repress from sites in the UAS. The yeast repressor Tup1 complexes with Ssn6 (also called Cyc8) with a ratio of 4:1, forming a complex of 420 kDa (*Varanasi et al., 1996*), and this complex represses transcription through a number of mechanisms. These include repositioning and stabilizing nucleosomes to form an inaccessible chromatin structure (*Chen et al., 2013*; *Zhang and Reese, 2004*; *Ducker, 2000*). Tup1 also blocks chromatin remodeling, masks activation domains, and excludes TBP (*Wong and Struhl, 2011*; *Zhang and Reese, 2004*; *Mennella et al., 2003*). LexA-Tup1 fusion proteins repress transcription when bound upstream of the Cyc1 promoter (*Tzamarias and Struhl, 1994*), and TetR-Tup1 fusions reduce uninduced expression in a dual TetR activator-repressor controller (*Bellí et al., 1998*). For $P_{7tet.1}$, we imagined that as many as seven TetR-Tup1 dimers might bind to the promoter, potentially recruiting two additional Tup1 and one Ssn6 molecules per $tetO_1$ site. The resulting ~3mDa of protein complexes might block activation by one or more of the above

mechanisms. We therefore measured the ability of a TetR-nls-Tup1 fusion to repress $P_{7tet.1}$-driven Citrine signal in SR strains. When its expression was directed from $P_{ACT1}$ (Y2669), TetR-nls-Tup1 decreased uninduced fluorescence signal to background levels (*Figure 3A*). Because fusion of TetR to a mammalian repressor domain in mammalian cells had shown very slow induction kinetics (*Deuschle et al., 1995*), we checked whether the TetR-nls-Tup1 fusion showed increased induction time compared to TetR alone but found no such effect (*Figure 3—figure supplement 1*). Additionally, TetR-nsl-Tup1 abolished uninduced expression driven by $P_{3tet}$ (*Figure 3—figure supplement 2*) (77-fold repression), compared to repression in otherwise isogenic strains by TetR, which showed basal expression reduced by only 1.5-fold (*Figure 1—figure supplement 1*). By contrast, TetR-nls-Tup1 fusion repressed $P_{2tet}$, where $tetO_1$ flank only the TATA sequence, more strongly than TetR alone, but still showed basal expression. Our data thus suggested that the TetR-nls-Tup1 suppressed basal expression mainly by its effects in the UAS (see Discussion).

In the cAR architecture, the induction threshold, that is, the smallest concentration of inducer that can induce expression, is determined by the number of molecules of the repressors present before induction. We sought to lower the induction threshold in order to maximize the input dynamic range. Therefore, we constructed cAR controllers using TetR and TetR-nls-Tup1, to determine the lowest level of TetR-nls-Tup1 that could still abolish uninduced expression from $P_{7tet.1}$. TetR-nls-Tup1 was driven by constitutive promoters of genes whose products were of decreasing abundance (*Ho et al., 2018*) ($P_{ACT1}$, $P_{VPH1}$, $P_{RNR2}$, $P_{REV1}$) (Y2673, Y2684, Y2749, and Y2715). The $P_{ACT1}$, $P_{VPH1}$

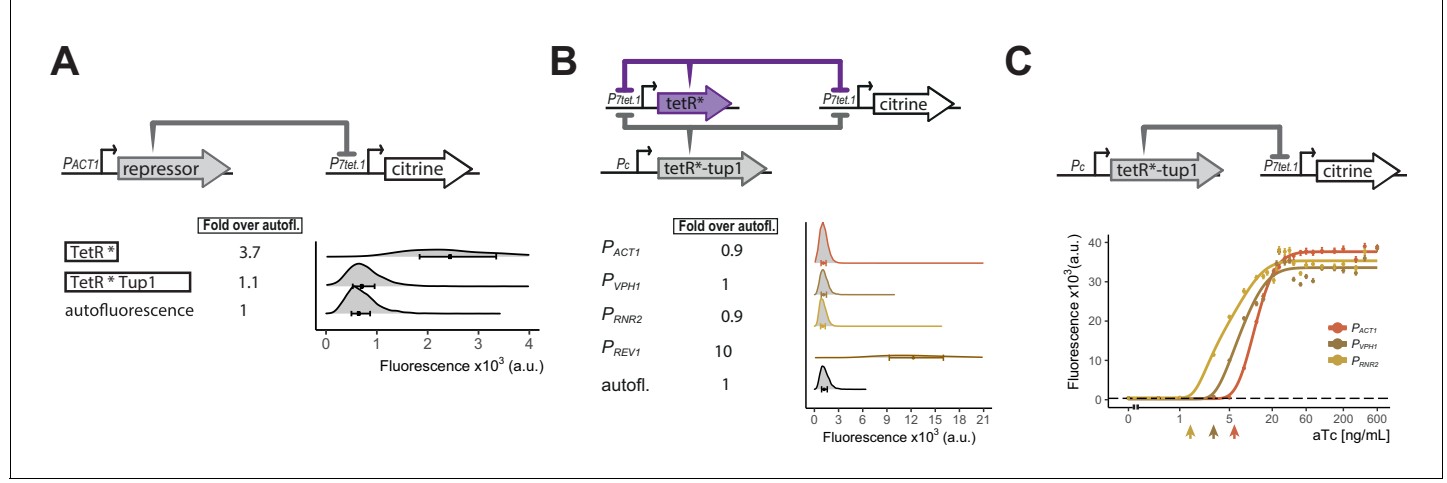

**Figure 3.** Repressor optimization to abolish $P_{7tet.1}$ basal expression. **A)** Testing repression by the TetR-Tup1 fusion. The top diagram indicates the genetic elements of the SR architecture used to test the ability of the TetR-Tup1 fusion to abolish basal expression from $P_{7tet.1}$. Diagrams to the left of the plot show the different repressors used. Each * indicates one SV40 Nuclear Localization Sequence. For both (**A**) and (**B**), Citrine fluorescence from $P_{7tet.1}$ repressed by the repressors indicated was measured using flow cytometry. Plots as in *Figure 1*. The circles inside each density plot show the median and the upper and lower bounds of the bar correspond to the first and third quartiles of the distribution. Numbers to the left of the plot indicate fold expression over autofluorescence, that is, the median of the Citrine fluorescence detected divided by the median of the autofluorescence signal. (**B**) Finding the lowest expression level of the zeroing repressor TetR-nls-Tup1 that abolishes basal expression from $P_{7tet.1}$. The top diagram shows the genetic elements of the cAR architecture in the strains tested. Pc indicates a constitutive promoter. Promoters driving TetR-nls-Tup1 expression are indicated to the left of the plot. Numbers to the left of the plot as in (**A**). (**C**) Reducing expression of TetR-nls-Tup1 lowers induction threshold. The top diagram shows genetic elements of SR architecture in which synthesis of TetR-nls-Tup1 was directed by different promoters. The plot shows Citrine fluorescence measured using flow cytometry at steady state, 7 hr after induction with different aTc concentrations. Arrows indicate induction thresholds, defined as the lowest aTc dose where an increase in fluorescence signal was detected. Dashed line indicates autofluorescence control (parent strain without Citrine), circles indicate the median of the experimentally measured population, lines are fitted. Error bars indicate 95% confidence interval calculated using bootstrapping (n=1000) as explained in Materials and methods.

The online version of this article includes the following source data and figure supplement(s) for figure 3:

**Source data 1.** Numerical data for *Figure 3*.

**Figure supplement 1.** The zeroing repressor TetR-nls-Tup1 does not affect the induction speed of $P_{7tet.1}$.

**Figure supplement 1—source data 1.** Numerical data for *Figure 3—figure supplement 1*.

**Figure supplement 2.** A TetR-nls-Tup1 fusion protein fully represses expression when binding only at the UAS, but not only at the TATA.

**Figure supplement 3.** Low level TetR-nls-Tup1 expression results in incomplete repression of $P_{7tet.1}$.

and $P_{RNR2}$ strains showed no uninduced expression (*Figure 3B*), while the $P_{REV1}$ strain did (*Figure 3—figure supplement 3*). Out of the three, Rrn2 protein is present at lower abundance, and the $P_{RNR2}$-driven TetR-nls-Tup1 has the lowest induction threshold in a dose response experiment with strains bearing SR architectures (Y2669, 2676, 2717) (*Figure 3C*).

We therefore chose as our final controller the cAR architecture in which $P_{7tet.1}$ directed the expression of both TetR and of the controlled gene, while $P_{RNR2}$ directed the synthesis of TetR-nls-Tup1. We constructed plasmids such that the tetR and tetR-nls-tup1 components are encoded on a single integrative plasmid, and a separate plasmid can be used to generate PCR fragments bearing $P_{7tet.1}$ for homologous recombination directed replacement of the promoter of any yeast gene. Due to its ability to give precisely regulated expression over a wide range of inducer concentrations, we called this construct a 'Well Tempered Controller' and gave it the number of Bach's first Prelude and Fugue (Bach, Johann Sebastian, 1685-1750. The Well Tempered Clavier. Book I: 24 Preludes and Fugues, BWV 846, C Maj) _ (*Figure 4A*).

## WTC$_{846}$ fulfills the criteria of an ideal transcriptional controller

We measured the time-dependent dose response of fluorescent signal in Y2759, the WTC$_{846}$::*citrine* strain during exponential growth using flow cytometry (*Figure 4B&C*). Without aTc, there was no signal above background. After induction, signal appeared within 30 min. Time to reach steady state, which will be shorter for proteins that degrade more quickly (see Appendix 4), was 7 hr for the stable protein Citrine. Steady state expression was adjustable over aTc concentrations from 0.5 ng/mL to 600 ng/mL, a 1200-fold input dynamic range. Maximum expression was similar to that for the $P_{TDH3}$-*citrine* strain Y2683. Direct observation of Citrine and TetR expression by Western blotting showed no expression of Citrine in absence of aTc, adjustable Citrine levels over the same input dynamic range and TetR expression synchronized with Citrine (*Figure 4—figure supplement 1*). In all eight growth media tested, WTC$_{846}$::*citrine* expression was precisely adjustable (*Figure 4—figure supplement 2*), and even very high induction of the WTC$_{846}$ system in a strain where only the control plasmid bearing tetR and tetR-nls-tup1 was integrated (Y2761) had no significant effect on growth rates (*Figure 4—figure supplement 3*).

To better characterize the system, we also measured the shutoff speed of WTC$_{846}$ driven expression. We reasoned that the time to observable phenotypic effect of WTC$_{846}$ shutoff would depend on the speed of five processes: (i) aTc diffusion out of the cell, (ii) TetR binding to its operators, (iii) aTc sequestration by newly synthesized TetR, (iv) degradation and dilution of citrine mRNA, and (v) degradation and dilution of Citrine protein. Processes i-iii would lead to the cessation of transcript production by WTC$_{846}$, and their speed would be the same for all WTC$_{846}$ controlled genes, whereas the speed of processes iv and v determine perdurance of the gene product and will be different for different mRNAs and proteins.

To measure shutoff, we grew the WTC$_{846}$::*citrine* strain (Y2759) to early exponential phase and then induced with a high concentration (600 ng/mL) of aTc and measured fluorescence signal every 30 min in flow cytometry. Additionally, after 30, 90, 150, and 210 min, we removed, washed, and resuspended a sample in (a) medium without aTc (to shut off expression from WTC$_{846}$) and (b) medium without aTc but with cyclohexamide (to shut off both WTC$_{846}$ and new protein synthesis). After shutoff, we expected to see an initial increase in signal, followed by decline from this peak. Increase in fluorescence after shutoff in (a) would depend on the time it took for WTC$_{846}$ to stop producing new mRNA (processes i-iii), the time it took for the existing mRNA to be degraded (process iv), and on continued fluorophore formation by already synthesized but immature Citrine proteins, which has a maturation time of around 30 min (*Nagai et al., 2002*). Whatever increase in fluorescence in (a) observed above that baseline found after shutoff in cycloheximide (b) would be due to WTC$_{846}$ shutoff speed and mRNA degradation speed.

As expected, we observed an initial increase of fluorescence in the shutoff samples (*Figure 4—figure supplement 8A*), which peaked for the samples in (a) (without cyclohexamide) at around 60 min. A single-cell division takes 90 min and we therefore conclude that WTC$_{846}$ shutoff (events i-iii) is rapid and occurs within one cell division, and likely within 30 min given that the time between Citrine production and observable fluorescence is around 30 min. Subsequent reduction in fluorescence, which fell to half after 120 min in all samples, is an estimation of process v, that is, Citrine degradation + dilution (*Figure 4—figure supplement 8B*), and is consistent with the idea that the continued reduction in Citrine signal is caused by dilution by cell division. Overall, we conclude that WTC$_{846}$

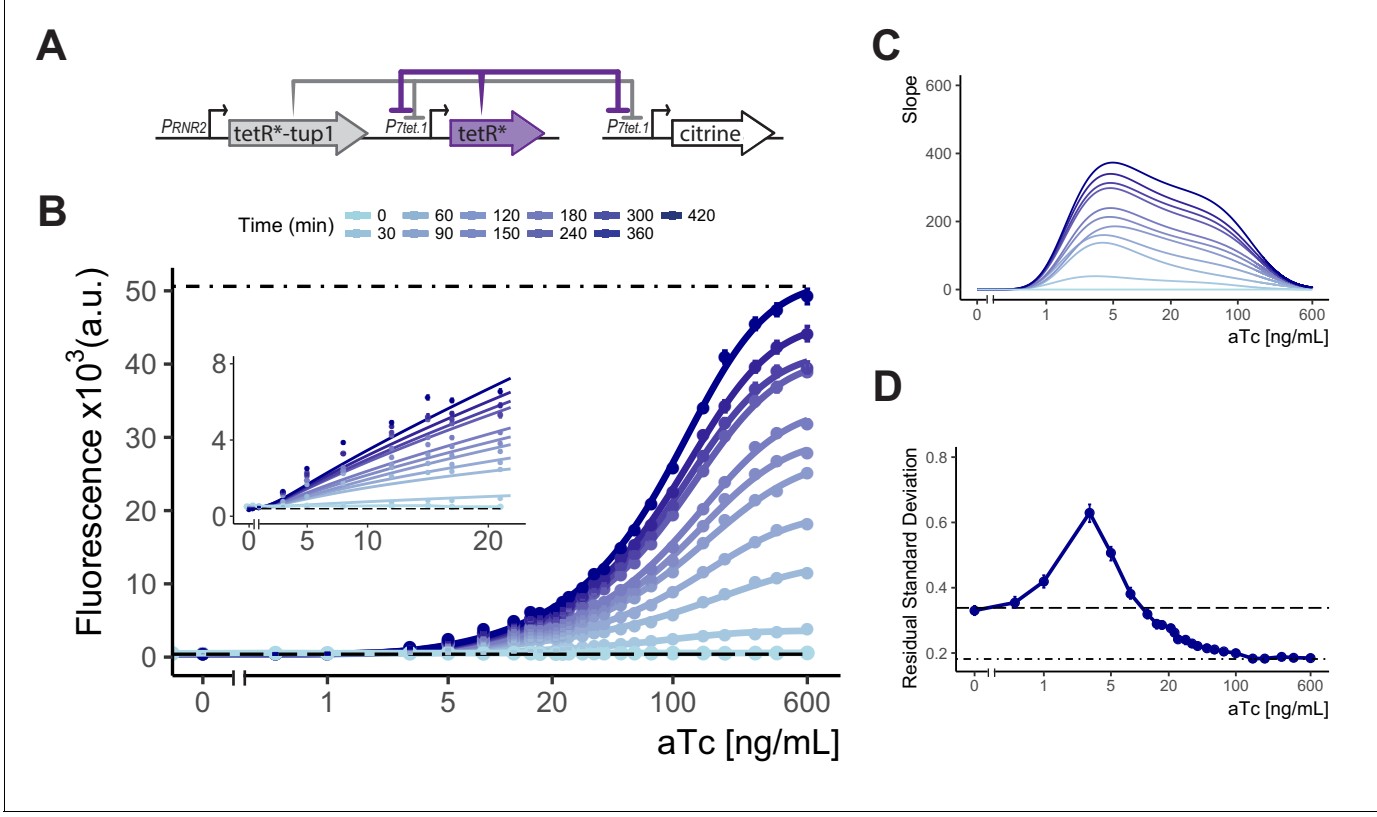

**Figure 4—source data 1.** Controlled gene expression from WTC$_{846}$. (**A**) Architecture of WTC$_{846}$. The final WTC$_{846}$ system is composed of a single integrative plasmid bearing TetR and TetR-Tup1 driven by the promoters indicated. This plasmid was integrated at the *URA3* locus. $P_{7tet.1}$-driven Citrine was integrated at the *LEU2* locus. * indicates SV40 Nuclear Localization Sequence. Repression of promoters is indicated by flat headed arrows. (**B**) Time dependent dose response of WTC$_{846}$-controlled expression. Citrine fluorescence was measured using flow cytometry at 30 min intervals after induction with different concentrations of aTc (ng/mL). Dashed line indicates median autofluorescence (parent strain without Citrine) and dot dashed line fluorescent signal from wild type $P_{TDH3}$ (Y2683). Circles show the median of the experimentally measured population, and the lines were fitted as explained in *Figure 2B*. The inset shows response at low input aTc doses. (**C**) The slopes of the dose response curves in (**A**), as a visual representation of the input dynamic range, defined as the range of doses where the slope of the dose response curve is non-zero. (**D**) Cell-to-cell variation of WTC$_{846}$-controlled expression. Single reporter CCV quantified using the VIV measure at 7 hr calculated as in 2D. Dashed line shows VIV of autofluorescence, dot-dashed line VIV of $P_{TDH3}$-driven Citrine signal. Where present, error bars indicate 95% confidence interval calculated using bootstrapping (n=1000) as described in Materials and methods.

The online version of this article includes the following source data and figure supplement(s) for figure 4:

**Source data 1.** Numerical data for *Figure 4* panels B and D.

**Figure supplement 1.** Direct observation of dose response for WTC$_{846}$-controlled protein expression.

**Figure supplement 1—source data 1.** Raw and uncropped images for western blots in *Figure 4—figure supplement 1*.

**Figure supplement 2.** Dose response of WTC$_{846}$-controlled expression in cells grown in different media.

**Figure supplement 2—source data 1.** Numerical data for *Figure 4—figure supplement 2*.

**Figure supplement 3.** WTC$_{846}$-directed expression does not affect gross measures of cell physiology.

**Figure supplement 3—source data 1.** Raw data and raw plate images for *Figure 4—figure supplement 3*.

**Figure supplement 4.** Cell-to-cell variation of WTC$_{846}$-driven expression during induction.

**Figure supplement 4—source data 1.** Numerical data for *Figure 4—figure supplement 4*.

**Figure supplement 5.** Peak CCV in SR strains corresponds to higher doses at higher expression levels of TetR-nls-Tup1.

**Figure supplement 5—source data 1.** Numerical data for *Figure 4—figure supplement 5*.

**Figure supplement 6.** Fluorescence and volume of the WTC$_{846}$::*citrine* strain induced with different aTc concentrations.

**Figure supplement 7.** CCV of WTC$_{846}$-controlled expression in cells grown in different media.

**Figure supplement 8.** Shutoff of WTC$_{846}$ expression.

**Figure supplement 8—source data 1.** Numerical data for *Figure 4—figure supplement 8*.

**Figure supplement 9.** Comparison of cell-to-cell variation between WTC$_{846}$, a previously published, β-estradiol induced transcriptional control system, and expression driven by $P_{GAL1}$.

**Figure supplement 9—source data 1.** Numerical data for *Figure 4—figure supplement 9*.

shutoff is rapid, but the time required to see the phenotypic effects of the absence of the controlled gene product will primarily depend on the stability of the mRNA and expressed protein.

We also quantified the cell-to-cell variation in Citrine expression using the single reporter VIV measure for the WTC$_{846}$::*citrine* strain (Y2759) grown in YPD, and compared it to variation in a β-estradiol (LexA-hER-B112) activation based transcriptional control system we previously described, and the commonly used galactose activated P$_{GAL1}$ (*Figure 4D*, *Figure 4—figure supplement 4*, *Figure 4—figure supplement 6* and *Figure 4—figure supplement 9*). At increasing concentrations of aTc, VIV initially rose to 0.63 at 8 ng/mL, similar to the VIV measured for Citrine expression repressed by P$_{RNR2}$-driven TetR-nls-Tup1 in an SR strain (Y2717, RSD of 0.67, *Figure 4—figure supplement 5*). At higher aTc inputs, VIV rapidly dropped below that seen in Y70, an otherwise-isogenic autofluorescence control strain, and reached the same low level (0.18) observed for Citrine whose expression was driven by P$_{TDH3}$ (Y2683). Because the autofluorescence varied so greatly, absolute VIV for cells grown in different media could not be directly compared. However, under all growth conditions (*Figure 4—figure supplement 7*), VIV was highest at the similarly low concentrations of aTc and decreased at higher concentrations to the levels shown by the P$_{TDH3}$-*citrine* strain (*Figure 4—figure supplement 2*). We interpret the peak of VIV in the input dynamic range as arising from the fact that the WTC$_{846}$ architecture combines Simple Repression and Autorepression of the P$_{7tet.1}$-controlled gene (here, Citrine). At low concentrations of inducer, in the SR regime, most repression of P$_{7tet.1}$ was due to the constitutively expressed TetR-nls-Tup1, and the peak VIV was similar to that found for the strain where P$_{7tet.1}$ was repressed by constitutively expressed TetR-nls-Tup1 (*Figure 4—figure supplement 5* and see previous Results section). At higher concentrations of aTc, in the AR regime, P$_{7tet.1}$ is derepressed, the concentration of TetR and the ratio of TetR to TetR-nls-Tup1 is large. At these inducer concentrations, TetR controls its own synthesis and variation is suppressed by this negative feedback, resulting in much lower cell-to-cell variation throughout the dynamic range compared to routinely used transcriptional controllers. Taken together, these results indicated that WTC$_{846}$ fulfilled our initially stated criteria for an ideal conditional expression system.

## WTC$_{846}$ alleles allow precise control over protein dosage and cellular physiology

We then assessed the ability of WTC$_{846}$ to direct conditional expression of endogenous genes. We selected (i) genes that are essential for growth, but for which previously generated transcriptionally controlled alleles still formed colonies on solid medium (*CDC42*, *TOR2*, *PBR1*, *CDC20*) or continued to grow in liquid medium (*PMA1*) under uninduced conditions, (ii) essential genes for which existing transcriptionally controlled alleles did not show the expected overexpression phenotype (*IPL1*), or (iii) essential genes for which conditional expression alleles did not exist (*CDC28*) (*Mnaimneh et al., 2004*; *Yu et al., 2006*; *Dechant et al., 2014*). These genes encoded proteins with a variety of functions: stable (Cdc28) and unstable (Cdc20 and Cdc42) cell cycle regulators, a spindle assembly checkpoint kinase (Ipl1), a metabolic regulator (Tor2), a putative oxidoreducatase (Pbr1), and a high abundance membrane proton pump (Pma1). The encoded proteins spanned a range of abundance from ~1000 (Tor2 and Ipl1) to >50,000 (Pma1) molecules per cell (*Ho et al., 2018*).

We constructed strains in which WTC$_{846}$ controlled the expression of these genes. Before transformation the cells were grown in liquid medium containing aTc, and then plated on solid medium containing aTc (see Appendix 5 for a detailed protocol) (*Figure 5A* and *Table 1*, strains labeled WTC$_{846-Kx}$::*gene_name*). To make these strains, we integrated a single plasmid-borne TetR-nls-Tup1 and autorepressing TetR construct into the *LEU2* locus in a BY4741 background, and replaced sequences upstream of the ATG of the essential gene with a ~1940 bp casette carrying an antibiotic selection marker and P$_{7tet.1}$, without altering the sequence of the upstream gene or its terminator. In most cases we removed between 20 and 200 bp of the endogenous gene promoter. The cassette carried one of three different 15 bp translation initiation sequences (extended Kozak sequences; K1, K2, K3) as the last 15 bases before the ATG. These were designed to enable different levels of translation of the gene's mRNA (*Li et al., 2017*). The predicted efficiency of the sequences was K1> K2> K3. If cells of a strain carrying a WTC$_{846}$-controlled essential gene formed colonies on solid medium without aTc, we constructed an otherwise-isogenic strain with a lower efficiency Kozak sequence (data not shown).

We spotted serial dilutions of cultures of the final seven strains on YPD, YPE, SD, S Glycerol and SD Proline plates with and without inducer, and assessed the strains' ability to grow into visible

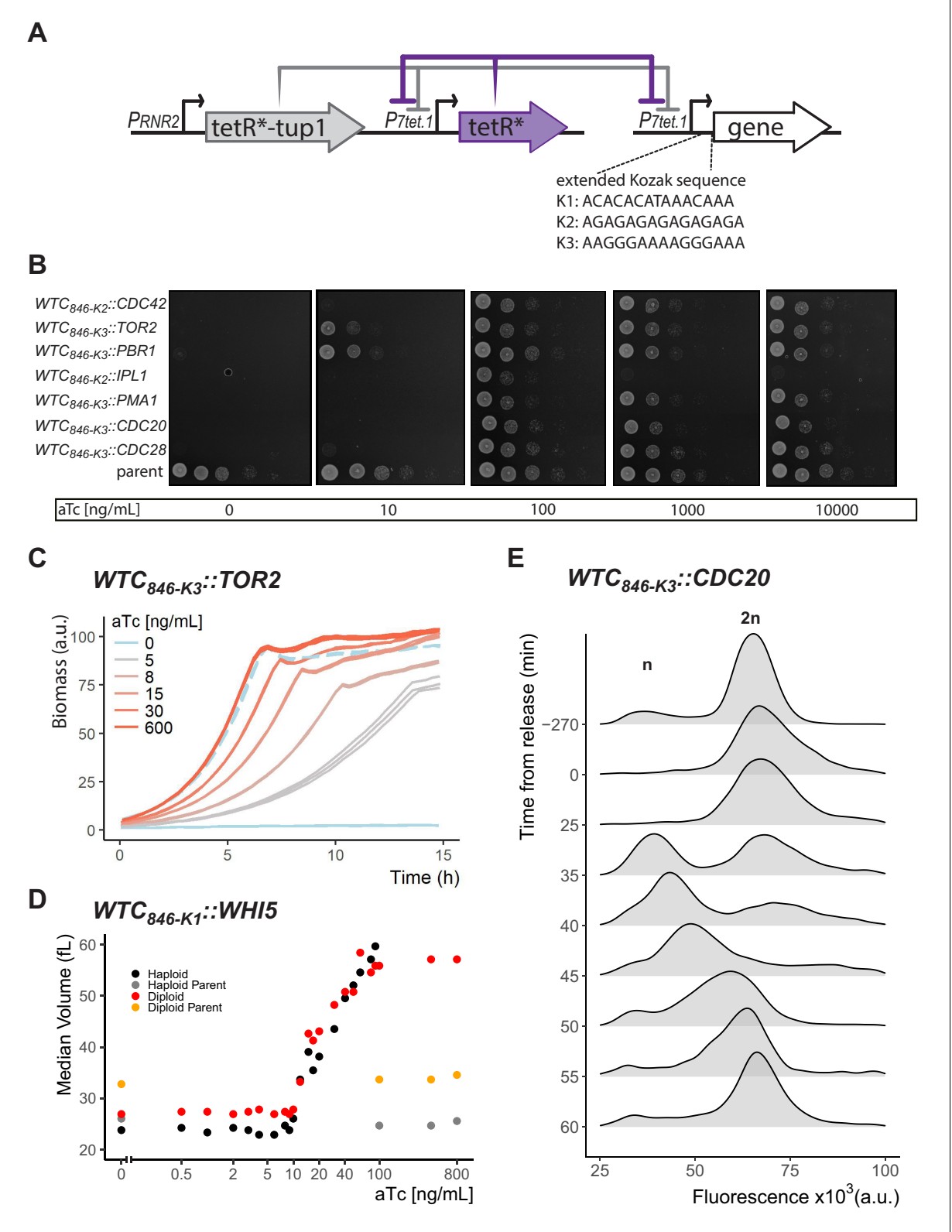

**Figure 5.** Controlled protein dosage of WTC$_{846}$-driven yeast genes. (A) The WTC$_{846}$ architecture used, as in *Figure 4A*. Figure also shows the three extended Kozak sequences used to control translation efficiency. (B) WTC$_{846}$ alleles of essential yeast genes show null and quantitative expression phenotypes. The genes whose expression is controlled by WTC$_{846}$ are indicated on the left. Cells growing in liquid medium were spotted onto different YPD plates, such that the leftmost circle on each plate had 2.25x10$^6$ cells and each subsequent column is a 1:10 dilution. aTc concentration in each

*Figure 5 continued*

plate is indicated below each image. Parent refers to the strain where all components of WTC$_{846}$ except the $P_{7tet.1}$ that directs expression of the controlled gene was present (Y2769). (**C**) Precise control of growth rate by adjusting Tor2 protein dosage. Growth of the WTC$_{846-K3}$::*TOR2* strain was measured by scattered light intensity using a growth reader. Cells were grown in liquid YPD, three replicate wells per aTc concentration were measured. Dashed line indicates the growth curve of the parent strain, where Tor2 was under endogenous control. The y-axis was normalized to a range between 0 and 100 and indicates culture density. (**D**) Precise control of cell volume by titrating dosage of Whi5. Haploid and Diploid refer to WTC$_{846-K1}$::*WHI5* alleles grown in S Ethanol with varying concentrations of aTc. Haploid and diploid parent indicates strains where Whi5 was under endogenous control. Median cell volume was measured using a Coulter Counter. (**E**) Batch culture cell cycle synchronization. A batch culture of WTC$_{846-K3}$::*CDC20* strain growing in 20 ng/mL aTc was arrested and synchronized by aTc withdrawal. Cells were released from the cell cycle block by addition of aTc at time 0. Cells were stained with Sytox and analyzed with flow cytometry. A total of 10,000 cells per time point were recorded. The plots are density distributions of the Sytox fluorescent signal of the whole population, such that the area under the curve equals 1. The peaks corresponding to one and two sets of chromosomes are labeled. These indicate the cells that are in G1 and G2/M phases of the cell cycle, respectively.

The online version of this article includes the following source data and figure supplement(s) for figure 5:

**Source data 1.** Raw plate images for *Figure 5B* and raw data for *Figure 5C*.

**Source data 2.** Numerical data for *Figure 5D*.

**Figure supplement 1.** Regulated protein dosage from WTC$_{846}$ alleles controls growth on different solid media.

**Figure supplement 1—source data 1.** Raw plate images for *Figure 5—figure supplement 1*.

**Figure supplement 2.** WTC$_{846}$-driven overexpression of Ipl1 prolongs G2/M and produces cells with >2 n ploidy.

**Figure supplement 3.** Adjustable protein dosage from WTC$_{846}$ alleles of essential and metabolic genes controls growth rates in different liquid media.

**Figure supplement 3—source data 1.** Raw data for *Figure 5—figure supplement 3*.

**Figure supplement 4.** Regulated clamped hypomorphic expression of WTC$_{846}$::*PMA1* allele causes cell separation defect.

**Figure supplement 5.** Whi5 titration leads to increased cell volume without an increase in cell-to-cell variation.

**Figure supplement 6.** Cell cycle arrest in the WTC$_{846-K3}$::*CDC20* strain.

**Figure supplement 7.** P$_{GAL1}$-driven expression from a centromeric plasmid results in high cell-to-cell variation.

colonies at a single time point, at which cells of the parent strain formed colonies in all serially diluted spots (24 hr for YPD and SD, 42 hr for others.) (*Figure 5B* and *Figure 5—figure supplement 1*). On all these media, no strain formed colonies without aTc and at intermediate concentrations of aTc all strains did. This result showed that WTC$_{846}$ alleles can produce null phenotypes.

At high aTc concentrations, the WTC$_{846-K2}$::*IPL1* strain formed colonies with lower plating efficiency than the parent strain. Ipl1 is a component of the kinetochore and is required for correct sister chromatid separation during mitosis. In mouse embryonic fibroblasts, overexpression of the ortholgous Aurora B kinase causes aberrant chromosome segregation and increases duration of mitosis by activating the Spindle Assembly Checkpoint, which stops mitosis until correct spindle attachments to sister chromatids can be formed (*González-Loyola et al., 2015*). In a previous study in *S. cerevisiae*, however, P$_{GAL1}$-driven overexpression of Ipl1 did not decrease plating efficiency, did not cause accumulation of cells with 2 n DNA content unable to complete mitosis, and did not cause aberrant chromosome segregation as assessed by microscopy, unless Ipl1 was overexpressed simultaneously with another kinetochore component (Sli15) (*Muñoz-Barrera and Monje-Casas, 2014*). We asked whether WTC$_{846}$-driven Ipl1 overexpression alone could cause missegregation phenotypes in *S. cerevisiae*. We cultured WTC$_{846-K2}$::*IPL1* cells for 18 hr in YPD with a high concentration of aTc (400 ng/mL), and measured total DNA content in flow cytometry to assess cell cycle state. In these cultures compared to the parent with WT Ipl1, many cells were in the G2/M phase with 2 n DNA content, indicative of an inability to complete mitosis, and a significant portion of the population showed aberrant chromosome numbers above 2 n (*Figure 5—figure supplement 2*). That is, WTC$_{846}$-driven Ipl1 overexpression in *S. cerevisiae* caused a previously undescribed phenotype, which resembled that caused by Aurora B overexpression in mammalian cells. To determine why WTC$_{846}$-driven Ipl1 overexpression caused this phenotype while P$_{GAL1}$-driven overexpression did not, we compared WTC$_{846}$-driven Citrine expression with Citrine driven by P$_{GAL1}$ carried on a centromeric plasmid. Compared with WTC$_{846}$-driven expression, centromeric P$_{GAL1}$ plasmid expression was twofold lower, and cell-to-cell variation was ~4.5-fold higher (*Figure 5—figure supplement 7*). Either the lower expression or the higher variation, or both, might account for the fact that P$_{GAL1}$ driven Ipl1 overexpression does not result in the mammalian Aurora B phenotype in *S. cerevisiae*.

We tested whether adjustable expression of metabolic and essential genes could be used to titrate growth rates. We constructed strains with WTC$_{846}$ alleles of Tor2, a low abundance, stable, essential protein necessary for nutrient signalling and actin polarization (*Bartlett and Kim, 2014*),

**Table 1.** Main strains used in this work and their relevant genotype.

A detailed table including all strains used in the supplementary figures can be found in the Supplement.

| Y | Name | Relevant genotype |
|---|------|-------------------|
| 70 | autofluorescence | BY4743 derivative, haploid, MATa *his3Δ1 leu2Δ0 met15Δ0 ura3Δ0 lys2Δ0* |
| 2683 | $P_{TDH3}$-const | *leu2Δ::$P_{TDH3}$_citrine-LEU2* |
| 2551 | $P_{2tet}$-const | *leu2Δ::$P_{2tet}$_citrine-LEU2* |
| 2564 | $P_{3tet}$-const | *leu2Δ::$P_{3tet}$_citrine-LEU2* |
| 2566 | $P_{5tet}$-const | *leu2Δ::$P_{5tet}$_citrine-LEU2* |
| 2562 | $P_{2tet}$-SR | *leu2Δ::$P_{2tet}$_citrine-LEU2 his3Δ::$P_{ACT1}$_tetR-NLS-HIS3* |
| 2573 | $P_{3tet}$-SR | *leu2Δ::$P_{3tet}$_citrine-LEU2 his3Δ::$P_{ACT1}$_tetR-NLS-HIS3* |
| 2577 | $P_{5tet}$-SR | *leu2Δ::$P_{5tet}$_citrine-LEU2 his3Δ::$P_{ACT1}$_tetR-NLS-HIS3* |
| 2659 | $P_{5tet.1}$-const | *leu2Δ::$P_{5tet.1}$_citrine-LEU2* |
| 2656 | $P_{5tet.1}$-SR | *leu2Δ::$P_{5tet.1}$_citrine-LEU2 his3Δ::$P_{ACT1}$_tetR-NLS-HIS3* |
| 2661 | $P_{7tet.1}$-const | *leu2Δ::$P_{7tet.1}$_citrine-LEU2* |
| 2663 | $P_{7tet.1}$-SR | *leu2Δ::$P_{7tet.1}$_citrine-LEU2 his3Δ::$P_{ACT1}$_tetR-NLS-HIS3* |
| 2674 | $P_{7tet.1}$-AR | *leu2Δ::$P_{7tet.1}$_citrine-LEU2 met15Δ::$P_{7tet.1}$_tetR-NLS-MET15* |
| 2741 | $P_{7tet.1}$-cAR | *leu2Δ::$P_{7tet.1}$_citrine-LEU2 met15Δ::$P_{7tet.1}$_tetR-NLS-MET15 his3Δ::$P_{ACT1}$_tetR-NLS-HIS3* |
| 2673 | $P_{7tet.1}$-cAR($P_{ACT1}$-TUP1) | *leu2Δ::$P_{7tet.1}$_citrine-LEU2 his3Δ::$P_{ACT1}$_tetR-NLS-tup1-HIS3 met15Δ::$P_{7tet.1}$_tetR-NLS-MET15* |
| 2684 | $P_{7tet.1}$-cAR($P_{VPH1}$-TUP1) | *leu2Δ::$P_{7tet.1}$_citrine-LEU2 his3Δ::$P_{VPH1}$_tetR-NLS-tup1-HIS3 met15Δ::$P_{7tet.1}$_tetR-NLS-MET15* |
| 2749 | $P_{7tet.1}$-cAR($P_{RNR2}$-TUP1) | *leu2Δ::$P_{7tet.1}$_citrine-LEU2 his3Δ::$P_{RNR2}$_tetR-NLS-tup1-HIS3 met15Δ::$P_{7tet.1}$_tetR-NLS-MET15* |
| 2715 | $P_{7tet.1}$-cAR(P_$P_{REV1}$-TUP1) | *leu2Δ::$P_{7tet.1}$_citrine-LEU2 his3Δ::P_$P_{REV1}$_tetR-NLS-tup1-HIS3 met15Δ::$P_{7tet.1}$_tetR-NLS-MET15* |
| 2669 | $P_{7tet.1}$-SR($P_{ACT1}$-TUP1) | *leu2Δ::$P_{7tet.1}$_citrine-LEU2 his3Δ::$P_{ACT1}$_tetR-NLS-tup1-HIS3* |
| 2676 | $P_{7tet.1}$-SR($P_{VPH1}$-TUP1) | *leu2Δ::$P_{7tet.1}$_citrine-LEU2 his3Δ::$P_{VPH1}$_tetR-NLS-tup1-HIS3* |
| 2717 | $P_{7tet.1}$-SR($P_{RNR2}$-TUP1) | *leu2Δ::$P_{7tet.1}$_citrine-LEU2 his3Δ::$P_{RNR2}$_tetR-NLS-tup1-HIS3* |
| 2759 | WTC$_{846}$::citrine | *leu2Δ::$P_{7tet.1}$_citrine-LEU2 ura3Δ::$P_{RNR2}$_tetR-NLS-tup1_$P_{7tet.1}$_tetR-NLS-URA3* |
| 2761 | WTC$_{846}$:: | *ura3Δ::$P_{RNR2}$_tetR-NLS-tup1_$P_{7tet.1}$_tetR-NLS-URA3* |
| 2769 | parent | *whi5Δ::WHI5-mKOkappa-HIS3, myo1Δ::MYO1-mKate(3x)-KanMX, leu2Δ::$P_{RNR2}$_tetR-NLS-tup1_$P_{7tet.1}$_tetR-NLS-LEU2* |
| 2772 | WTC$_{846-K1}$::TOR2 | *whi5Δ::WHI5-mKOkappa-HIS3, myo1Δ::MYO1-mKate(3x)-KanMX, leu2Δ::$P_{RNR2}$_tetR-NLS-tup1_$P_{7tet.1}$_tetR-NLS-LEU2 P_TOR2::$P_{7tet.1-K1}$-HygMX* |
| 2775 | WTC$_{846-K2}$::CDC28 | *whi5Δ::WHI5-mKOkappa-HIS3, myo1Δ::MYO1-mKate(3x)-KanMX, leu2Δ::$P_{RNR2}$_tetR-NLS-tup1_$P_{7tet.1}$_tetR-NLS-LEU2 P_CDC28::$P_{7tet.1-K2}$-NatMX* |
| 2837 | WTC$_{846-K3}$::CDC20 | *whi5Δ::WHI5-mKOkappa-HIS3, myo1Δ::MYO1-mKate(3x)-KanMX, leu2Δ::$P_{RNR2}$_tetR-NLS-tup1_$P_{7tet.1}$_tetR-NLS-LEU2 P_CDC20::$P_{7tet.1-K3}$-NatMX* |
| 2788 | WTC$_{846-K2}$::CDC42 | *whi5Δ::WHI5-mKOkappa-HIS3, myo1Δ::MYO1-mKate(3x)-KanMX, leu2Δ::$P_{RNR2}$_tetR-NLS-tup1_$P_{7tet.1}$_tetR-NLS-LEU2 P_CDC42::$P_{7tet.1-K2}$-NatMX* |
| 2789 | WTC$_{846-K2}$::IPL1 | *whi5Δ::WHI5-mKOkappa-HIS3, myo1Δ::MYO1-mKate(3x)-KanMX, leu2Δ::$P_{RNR2}$_tetR-NLS-tup1_$P_{7tet.1}$_tetR-NLS-LEU2 P_IPL1::$P_{7tet.1-K2}$-NatMX* |
| 2828 | WTC$_{846-K3}$::PMA1 | *whi5Δ::WHI5-mKOkappa-HIS3, myo1Δ::MYO1-mKate(3x)-KanMX, leu2Δ::$P_{RNR2}$_tetR-NLS-tup1_$P_{7tet.1}$_tetR-NLS-LEU2 P_PMA1::$P_{7tet.1-K3}$-NatMX* |
| 2773 | WTC$_{846-K3}$::TOR2 | *whi5Δ::WHI5-mKOkappa-HIS3, myo1Δ::MYO1-mKate(3x)-KanMX, leu2Δ::$P_{RNR2}$_tetR-NLS-tup1_$P_{7tet.1}$_tetR-NLS-LEU2 P_TOR2::$P_{7tet.1-K3}$-HygMX* |
| 2827 | WTC$_{846}^{-K3}$::CDC28 | *whi5Δ::WHI5-mKOkappa-HIS3, myo1Δ::MYO1-mKate(3x)-KanMX, leu2Δ::$P_{RNR2}$_tetR-NLS-tup1_$P_{7tet.1}$_tetR-NLS-LEU2 P_CDC28::$P_{7tet.1-K3}$-NatMX* |
| 2830 | WTC$_{846-K3}$::PBR1 | *whi5Δ::WHI5-mKOkappa-HIS3,myo1Δ::MYO1-mKate(3x)-KanMX, leu2Δ::$P_{RNR2}$_tetR-NLS-tup1_$P_{7tet.1}$_tetR-NLS-LEU2 P_PBR1::$P_{7tet.1-K3}$-NatMX* |

*Table 1 continued on next page*

Table 1 continued

| Y | Name | Relevant genotype |
| --- | --- | --- |
| 2849 | WTC$_{846-K3}$::TPI1 | leu2Δ::P$_{RNR2}$_tetR-NLS-tup1_P$_{7tet.1}$_tetR-NLS-LEU2 P_TPI1::P$_{7tet.1-K3}$-NatMX |
| 2791 | WTC$_{846-K1}$::WHI5 | whi5Δ::WHI5-mKOkappa-HIS3, myo1Δ::MYO1-mKate(3x)-KanMX, leu2Δ::P$_{RNR2}$_tetR-NLS-tup1_P$_{7tet.1}$_tetR-NLS-LEU2 P_WHI5::P$_{7tet.1-K1}$-NatMX |
| 2929 | WTC$_{846-K1}$::WHI5 (diploid) | BY4743, whi5Δ::WHI5-mKokappa-HIS3/WHI5 myo1Δ::MYO1-mKate(3x)-KanMX/MYO1 leu2Δ::P$_{RNR2}$_tetR-NLS-tup1_P$_{7tet.1}$_tetR-NLS-LEU2/leu2Δ0 ura3Δ::P$_{RNR2}$_tetR-NLS-tup1_P$_{7tet.1}$_tetR-NLS-URA3/ura3Δ0 P_WHI5::P$_{7tet.1-K1}$-HygMX/ P_WHI5::P$_{7tet.1-K1}$-NatMX |

Pma1, an abundant, essential proton pump that regulates the internal pH of the cell (*Ambesi et al., 2000*), and Tpi1, a highly abundant, non-essential glycolytic enzyme (*Fraenkel, 2003*) (Y2773, 2828, 2849). We cultured WTC$_{846}$::TOR2, WTC$_{846}$::PMA1, and WTC$_{846}$::TPI1 cells in different liquid media over a large input dynamic range of aTc, and measured growth by scattered light intensity in a growth reader as a proxy for culture density (Biolector or GrowthProfiler) (*Figure 5C* for Tor2 and *Figure 5—figure supplement 3* for all three proteins). All strains showed distinct growth rates at different aTc concentrations. For all strains, we identified an aTc concentration that resulted in the same growth rate as the otherwise-isogenic strain bearing the native gene promoter. In order to assess whether the WTC$_{846}$::PMA1 strain showed the expected hypomorphic phenotype of defective daughter cell separation (*Cid et al., 1987*), we used flow cytometry and Sytox Green staining to quantify DNA content. At low aTc concentrations, cells showed an apparent increase in ploidy and cell size and microscopic observation showed that each mother had multiple daughters attached to it (*Figure 5—figure supplement 4*). Observation of WTC$_{846}$::TOR2 strains revealed a novel overexpression phenotype: at high aTc concentrations, cells bearing the higher translational efficiency TOR2 allele (WTC$_{846-K1}$::TOR2) grew more slowly than the otherwise-isogenic control parent strain with WT TOR2 (*Figure 5—figure supplement 3D*, compare 600 ng/mL line to blue dashed line). The strain with the less efficient WTC$_{846-K3}$::TOR2 allele did not show this overexpression phenotype. These results demonstrate that researchers can adjust input to WTC$_{846}$ alleles to tune protein levels and different growth rates with a level of precision not achievable until now, and that the dynamic range of phenotypic outputs can be further expanded by the ability to construct WTC$_{846}$ alleles with alternative Kozak sequences to observe phenotypes at the two dosage extremes.

We then tested whether adjustable gene expression could precisely regulate cell size. In *S. cerevisiae*, Whi5 regulates the volume at which unbudded cells commit to a round of division and start forming buds. whi5Δ cells are smaller, and cells expressing Whi5 under P$_{GAL1}$ control are larger than otherwise-isogenic cells (*de Bruin et al., 2004*). Whi5 controls cell volume by a complex mechanism and unlike most other proteins, its abundance does not scale with cell volume (*Schmoller et al., 2015*). Whi5 mRNA and protein are expressed during S/G2/M (in haploids, at about 2500 molecules), and Whi5 is imported into the nucleus in late M phase (*Taberner et al., 2009*), where it suppresses transcription of the G1 cyclins needed to commence a new round of cell division (*de Bruin et al., 2004*; *Taberner et al., 2009*). During G1, as cells increase in volume, the nuclear concentration of Whi5 falls due to dilution (*Schmoller et al., 2015*) and slow nuclear export (*Qu et al., 2019*) until a threshold is reached, after which Whi5 is rapidly exported from the nucleus, and cells enter S phase. To test whether we could control cell volume by controlling Whi5, we constructed haploid and diploid WTC$_{846}$::WHI5 strains (Y2791, Y2929). In these strains, we expected Whi5 to be expressed throughout the cell cycle, but that import of the protein into the nucleus during late M phase, and diminution of nuclear concentration to below the threshold needed to START as cell volume increased in G1, should remain unaffected. We expected that the volume of these cells should scale with the concentration of the aTc inducer. We grew these strains along with otherwise isogenic control strains in S Ethanol to exponential phase at different aTc concentrations, and measured cell volume using a Coulter counter. Increasing Whi5 expression resulted in increasingly larger cells (*Figure 5D*). Without aTc, diploid WTC$_{846}$::WHI5 cells were about the same volume as haploid controls (median 27fL vs 25fL), whereas haploid WTC$_{846}$::WHI5 cells were only slightly smaller at 24fL. At around 10 and 12 ng/mL aTc, both haploid and diploid strains had about the same volume as controls. At full induction, both WTC$_{846}$::WHI5 strains had a median volume of around 60fL, almost twice as large as the diploid control, yielding a more than twofold range of possible cell volumes

attainable using WTC$_{846}$ for both haploid and diploid cells. We also calculated the CoV of cell volume to assess cell-to-cell variation of this WTC$_{846}$ directed phenotype. For most of the volume range, the CoV was around the same level as for the control strains with WT Whi5 (*Figure 5—figure supplement 5A&B*). Both diploid and haploid cells (especially haploids) expressing high levels of Whi5 showed increased variation in volume. We quantified DNA content of the haploid strain in a high aTc concentration using Sytox staining and found an increase in the number of aneuploid cells (>2n) (*Figure 5—figure supplement 5C*). We therefore believe that overexpression of Whi5 leads to endoreplication, and the increased variation in volume at high aTc concentrations in the haploid strain originates from these endoreplicated cells.

Finally, we tested the ability of WTC$_{846}$ to exert dynamic control of gene expression by constructing a WTC$_{846-K3}$::*CDC20* strain (Y2837) and using this allele to synchronize cells in batch culture by setting Cdc20 expression to zero and then restoring it (*Juanes, 2017*). Cdc20 is an essential activator of Anaphase Promoting Complex C, which once bound to Cdc20, initiates the mitotic metaphase to anaphase transition (*Pesin and Orr-Weaver, 2008*), and is then degraded during anaphase. Upon depletion of Cdc20, for example by shift of *ts* strains to the restrictive temperature, or transcriptionally controlled alleles to non-inducing medium, cells arrest in metaphase with large buds and 2 n DNA content. When Cdc20 is restored by switching to the permissive condition, cells enter the next cell cycle simultaneously (*Cosma et al., 1999*; *Shirayama et al., 1998*). For an investigator to be able to use WTC$_{846-K3}$::*CDC20* to synchronize the cells in a culture, the investigator would need to shut off Cdc20 expression completely, and then re-express it in all the cells in a population. To test the feasibility of this, we diluted exponentially growing WTC$_{846-K3}$::*CDC20* cells into YPD medium without aTc (0.5 million cells/mL) and took samples for Sytox staining and flow cytometry analysis for DNA content at fixed intervals. Within 480 min, the entire culture had arrested at the G2/M phase with 2 n DNA content (*Figure 5E*). Microscopic inspection confirmed that cells had arrested with large buds, as is expected upon a G2/M arrest. We next added 600 ng/mL of aTc. As assayed by Sytox staining and flow cytometry and confirmed by microscopy, cells then re-entered the cell cycle within 35 min and went through one cell cycle completely synchronously. Induction of WTC$_{846}$ is thus rapid, indicating that diffusion of aTc into the cell, and TetR unbinding of *tetO* are also rapid.

We also determined the arrest time of cells pre-cultured with a lower concentration (3 ng/mL) of aTc (*Figure 5—figure supplement 6*). These cells had a lower concentration of Cdc20 before aTc was removed, and therefore required less time to reach complete arrest (~210 min as opposed to ~480 min). This suggests that the predominant contribution to the time to reach complete arrest is the concentration and stability of Cdc20. Given this, and the rapid shutoff kinetics of WTC$_{846}$ presented in *Figure 4—figure supplement 8*, we conclude that the shutoff dynamics of WTC$_{846}$ controlled phenotypes depend mostly on the speed of degradation of the controlled protein. Additionally, when compared to published data (*Tavormina and Burke, 1998*; *Cosma et al., 1999*; *Ewald et al., 2016*), arrest at G2/M using the WTC$_{846-K3}$::*CDC20* strain is more penetrant than that obtained using temperature-sensitive (~25% unbudded cells) and transcriptionally controlled (~10% unbudded) alleles of *CDC20*. Release is at least just as fast as that observed for the temperature-sensitive (~35 min) and the transcriptionally controlled allele (~40 min).

## Discussion

Conditional expression of genes and observation of phenotype remain central to biological discovery. Many methods used historically, such as suppression of nonsense mutations, or conditional inactivation of temperature sensitive mutations, do not facilitate titration of graded or intermediate doses of protein. More current methods for graded expression do not allow experimenters to adjust and set protein levels and show high cell-to-cell variation of protein expression in cell populations, limiting their utility for elucidating protein-dosage-dependent phenotypes. Moreover, most such methods also have secondary consequences including slowing of cell growth. In order to overcome these limitations, we developed for use in *S. cerevisiae* a 'Well-tempered Controller'. This controller, WTC$_{846}$, is an autorepression-based transcriptional controller of gene expression. It can set protein levels across a large input and output dynamic range. As assessed by Citrine fluorescence readout, WTC$_{846}$ alleles display no uninduced basal expression, and uninduced WTC$_{846}$ alleles of poorly expressed proteins display complete null phenotypes. WTC$_{846}$ alleles also exhibit high maximum

expression, low cell-to-cell variation, and operation in different media conditions without adverse effects on cell physiology.

The central component of $WTC_{846}$ is an engineered TATA containing promoter, $P_{7tet.1}$. We and others had shown that prokaryotic repressors including LexA (**Brent and Ptashne, 1984**), TetR (**Murphy et al., 2007**), λcI (**Wedler and Wambutt, 1995**) and LacI (**Hu and Davidson, 1987**; **Figge et al., 1988**) can block transcription from engineered TATA-containing eukaryotic promoters, when those promoters contain binding sites between the UAS (or, for vertebrate cells, the enhancer) and the TATA (**Brent and Ptashne, 1984**) or downstream of or flanking the TATA. To develop $P_{7tet.1}$, we placed seven $tetO_1$ TetR-binding sites in the promoter of the strongly expressed yeast gene *TDH3*. Two of the sites flank the TATA sequence, the other five abut binding sites for an engineered UAS that binds the transcription activators Rap1 and Gcr1. In $WTC_{846}$, one instance of $P_{7tet.1}$ drives expression of the controlled gene, while a second instance of $P_{7tet.1}$ drives expression of the TetR repressor, which thus represses its own synthesis.

We believe that repression of $P_{7tet.1}$ by TetR is due mainly to its action at the two $tetO_1$ sites flanking the TATA sequence, because TetR represses a precursor promoter that only carries such sites to the same extent. The mechanism(s) by which binding of repressors near the TATA might interfere with PIC formation, transcription initiation, or early elongation remain unknown, as well as why binding of larger presumably transcriptionally inert TetR fusion proteins results in stronger repression. However, examination of the Cryo-EM structure of TBP and TFIID bound to mammalian TATA promoters (**Nogales et al., 2017**) suggests that binding of TetR and larger derivatives of it to these sites might simply block PIC assembly. Studies of repression of native *Drosophila melanogaster* promoters by the *en* and *eve* homeobox proteins show that a similar, steric occlusion based mechanism can block eukaryotic transcription by binding of the repressors to sites close to the TATA sequence (**Ohkuma et al., 1990**; **Austin and Biggin, 1995**).

In $WTC_{846}$, when inducer is absent, measured basal expression of the controlled gene is abolished by a second TetR derivative, a fusion bearing an active repressor protein native to yeast. Because the same TetR-nls-Tup1 fusion protein fully represses a precursor promoter that only carries TetR operators in the UAS, we believe that the main zeroing effect of TetR-nls-Tup1 is manifested through binding the $tetO_1$ sites in the UAS. Native Tup1 repressor complexes with Ssn6 (also called Cyc8) to form a 420 kDa protein complex (**Varanasi et al., 1996**), and TetR binds DNA as a dimer. In *gcr1Δ* cells, in which transcription from $P_{TDH3}$ is severely diminished, the native $P_{TDH3}$ promoter has two nucleosomes positioned between the UAS and the transcription start site (**Pavlović and Hörz, 1988**). It is thus possible that in the UAS as many as five very large dimeric TetR-nls-Tup1 complexes block binding of Gcr1 and Rap1, mask their activating domains, or some combination of these, resulting in similar placement of two nucleosomes in $P_{7tet.1}$. One of these nucleosomes could then be positioned at the 294nt stretch between the UAS and the TATA sequence. It also seems possible that binding of the TetR-nls-Tup1 repressor might shift the position of the second nucleosome further downstream, so that it obscures the transcription start site.

Both the increased input dynamic range and the lower cell-to-cell variation in expression from $WTC_{846}$ arise from the fact that the TetR protein that represses the controlled gene also represses its own synthesis. This autorepression architecture is common in prokaryotic regulons (**Smith and Magasanik, 1971**; **Brent and Ptashne, 1980**) including Tn10, the source of the TetR gene used here, and it has been engineered into eukaryotic systems (**Becskei and Serrano, 2000**; **Nevozhay et al., 2009**). In self-repressing TetR systems, the input (here, aTc) and TetR output together function as a comparator-adjustor (**Andrews et al., 2016**). In such systems, aTc diffuses into the cell. Intracellular aTc concentration is limited by entry. Inside the cell, aTc and $TetR^{free}$ concentrations are continuously compared by their binding interaction. If $TetR^{free}$ is in excess, it represses TetR expression, and total intracellular TetR concentration is reduced by dilution, cell division, and active degradation of DNA-bound TetR until an equilibrium determined by the intracellular aTc concentration is again reached. The consequence of this autorepression is that the $WTC_{846}$ requires more aTc to reach a given level of controlled gene expression than strains in which TetR is expressed only constitutively. Autorepression flattens the dose response curve, increases the range of aTc doses where a change in promoter activity can be observed, and buffers the effects of stochastic cell-to-cell variations in TetR concentration, thereby reducing cell-to-cell variation in expression of the controlled gene throughout the input dynamic range.

We further extended the output dynamic range of WTC$_{846}$-controlled genes by developing three different Kozak sequences, K1, K2, and K3 (*Li et al., 2017*), to allow controlled genes to be translated at different levels. We used these sequences to construct strains bearing conditional alleles of the essential genes *CDC28*, *IPL1*, *TOR2*, *CDC20*, *CDC42*, *PMA1*, and *PBR1*. These strains all showed graded expression of growth and other phenotypes, from lethality at zero expression to penetrant expression of previously reported phenotypes at higher protein dosage (*Mnaimneh et al., 2004*; *Yu et al., 2006*; *Dechant et al., 2014*; *Muñoz-Barrera and Monje-Casas, 2014*). We used controlled expression in the WTC$_{846}$::*CDC20* strain to bring about G2/M arrest followed by synchronous release, with low cell-to-cell variation in induction timing, demonstrating that WTC$_{846}$ can be used in experimental approaches that require dynamic control of gene expression.

Both induction and shutoff with WTC$_{846}$ are rapid, as both Citrine and Cdc20 expression occur within 30 min of induction, and shutoff of Citrine expression is observed within 60 min. However, time to steady state expression after induction, reached when degradation and dilution through cell division balance new synthesis, takes longer. Time to steady state will depend on the stability of the controlled protein. This is 6–7 hr for the stable protein Citrine, and the majority of yeast proteins have similar stability (*Wiechecki et al., 2018*). Those proteins with shorter half-lives will reach steady state faster. Furthermore, we showed in WTC$_{846}$::*IPL1* strains that high level expression of this spindle assembly checkpoint kinase arrests cells at G2/M with 2 n or higher DNA content. This phenotype, thought to be due to disruption of kinetichore microtubule attachments, is displayed in mammalian cells when the homologous Aurora B is overexpressed (*González-Loyola et al., 2015*), but had not been observed previously in *S. cerevisiae* when Ipl1 was overexpressed from $P_{GAL1}$ (*Muñoz-Barrera and Monje-Casas, 2014*). We also showed that in WTC$_{846}$::*WHI5* strains, different levels of controlled expression of Whi5 can constrain cell sizes within different limits.

Cell-cell variation in WTC$_{846}$-driven expression is highest at low aTc levels, because control in this regime depends mostly on the higher variation Simple Repression by TetR-Tup1 expressed from the $P_{RNR2}$. This variation at low input doses in the WTC$_{846}$represented a trade-off between the design goals of abolition of basal expression and suppression of cell-to-cell variation. The Autorepression (AR) architecture better suppressed cell-to-cell variation in controlled gene expression at low inducer inputs, but, because of the fact that TetR and the controlled gene were both under the control of the same repressible promoter, the controlled gene still showed considerable basal expression when uninduced. Given that suppression of basal expression of the controlled gene was critical to generating 'reversible null' phenotypes, we developed the AR architecture further. The resulting cAR configuration of WTC$_{846}$, had low cell-to-cell variation, equivalent to the variation at the lowest expression levels that AR could achieve. Importantly, because transcription of WTC$_{846}$-controlled genes is synchronized to that of the autorepressing TetR gene, transcription and mRNA abundance of WTC$_{846}$-controlled genes should be steady throughout the cell cycle. This autorepressing circuitry operationally defines WTC$_{846}$ as an 'expression clamp', a device for adjusting and setting gene expression at desired levels, and maintaining it with low cell-to-cell variation, and so allowing expressed protein dosage in individual cells to closely track the population average.

Taken together, our results show that WTC$_{846}$ controlled genes define a new type of conditional allele, one that allows precise control of gene dosage. We anticipate that WTC$_{846}$ alleles will find use in cell biological experimentation, for example in assessment of phenotypes now incompletely penetrant due to variable dosage of the causative gene products (*Casanueva et al., 2012*), and for sharpening the thresholds at which dosage dependent phenotypes manifest. We also hope that genome wide collections of WTC$_{846}$ alleles might enable genome wide gene-by-gene and gene-by-chemical epistasis for interactions that depend on gene dosage. In *S. cerevisiae*, recent development of strains and methods (such as the SWAP-Tag [*Weill et al., 2018*]) that facilitate installation of defined N and C terminal genetic elements after cycles of mating, sporulation, and selection of desired haploids should allow generation of whole genome WTC$_{846}$ strains for this purpose. Epistasis screens rely on measurement of colony size on plates or culture density in liquid media. For two proteins whose effect on growth rate was identical, a one-generation difference in achievement of steady state expression could result in a twofold difference in number of cells in a colony or well, and thus in a 1.26-fold difference in colony diameter. We therefore suggest that growth rate-based assays using WTC$_{846}$ or any other inducible system pre-induce cells several generations before plating or pinning. WTC$_{846}$ alleles may find use in engineering applications such affinity maturation of antibodies expressed by yeast surface display, where precise ability to lower surface concentration

should aid selection for progressively higher affinity binders. $WTC_{846}$ can also be a useful complement to boost the efficiency of methods that act at the protein level such as induced degradation or AnchorAway techniques. Such techniques could be used in conjunction with $WTC_{846}$ to achieve rapid and sustained shutoff from a well-maintained steady state level. This would also allow fast step function decreases in abundance. For example, an experimenter might simultaneously induce depletion of the product of a controlled gene by such a method while adjusting aTc downward to rapidly reset the level of an expressed protein to a new, lower level. Implementation of the $WTC_{846}$ control logic in mammalian cells and in engineered multicellular organisms should allow similar experimentation now impossible due to cell-to-cell variation and imprecise control.

## Materials and methods

### Plasmids

Information on plasmids, and promoter and protein sequences used in this study can be found in *Supplementary file 1* - Tables S2 and S4. Plasmids with auxotrophic markers were constructed based on the pRG shuttle vector series (*Gnügge et al., 2016*) using either restriction enzyme cloning or isothermal assembly (*Gibson et al., 2009*). Inserts were generated either by PCR on existing plasmids or custom DNA synthesis (GeneArt, UK). Oligos for cloning and for strain construction were synthesized by Thermofisher, UK. Plasmids used to generate linear PCR products for tagging transformations were based on the pFA6 backbone (*Janke et al., 2004*). Plasmids necessary to construct $WTC_{846}$ strains are available through Addgene. Plasmid structures and a detailed protocol for strain construction can be found in Appendix 5.

pRG shuttle vector series backbones used for integrative transformations have T7 and T3 promoters flanking the insert (*Gnügge et al., 2016*). During cloning, the insert of plasmids bearing TetR were cloned such that the insert promoter was closer to the T7 promoter and the terminator was near the T3 promoter of the backbone. In plasmids bearing Citrine, the insert was flipped onto the opposite strand, such that the insert promoter was near the T3 promoter, and the terminator near the T7 promoter. This inversion was done to avoid homologous recombination during subsequent integration of these plasmids into the same strain, since in many strains TetR and Citrine were flanked by the same promoter and the same terminator.

### Strains

Strains used in this study can be found in *Supplementary file 1* - Table S1. Strains used for fluorescent measurements and the $WTC_{846-K3}::TPI1$ strain are based on a BY4743 derivative haploid background (MATa *his3Δ leu2Δ met15Δ ura3Δ lys2Δ*). Strains where $P_{7tet.1}$ replaced endogenous promoters were based on the haploid BY4741 background with the modifications *whi5Δ::WHI5-mKOkappa-HIS3*, *myo1Δ::MYO1-mKate(3x)-KanMX* and so were resistant to G418. The oligos used to replace the promoters of the different endogenous genes with $WTC_{846}$-controlled $P_{7tet.1}$ can be found in *Supplementary file 1* - Table S3. Correct replacement of the endogenous promoter with $P_{7tet.1}$ was checked using colony PCR with the protocol from the Blackburn lab (also detailed in *Gnügge et al., 2016*), and subsequent sequencing (Microsynth, Switzerland). For colony PCR, we used a standard forward oligo annealing to $P_{7tet.1}$, and gene specific reverse oligos annealing within the tagged gene. Oligo sequences for colony PCR can be found in *Supplementary file 1* - Table S3. A comprehensive protocol on how to generate strains where $WTC_{846}$ controls endogenous genes can be found in Appendix 5.

### Chemicals and media

YPD/YPE was prepared with 1% yeast extract (Thermofisher, 212720), 2% bacto-peptone (Thermofisher, 211820), and 2% glucose (Sigma, G8270) / ethanol (Honeywell, 02860). Synthetic (S) media except SD Proline contained 0.17% yeast nitrogen base (without amino acids and ammonium sulfate) (BD Difco, 233520) with 0.5% ammonium sulfate (Sigma, 31119) as nitrogen source, complete complement of amino acids and adenine and uracil, except for SD min which contained only the necessary amino acid complements to cover auxotrophies. SD Proline media contained 0.17% yeast nitrogen (without amino acids and ammonium sulfate), only the amino acids necessary to cover auxotrophies and 1 mg/mL proline as the sole nitrogen source. The carbon source was 2% glucose for SD

and SD Proline, 2% ethanol for S Ethanol, 3% glycerol for S Glycerol (Applichem, A2957), 2% fructose for S Fructose, 2% Raffinose for S Raffinose and 2% Galactose together with 2% Raffinose for S GalRaf. Experiments were performed in YPD media unless otherwise specified. Solid medium plates were poured by adding 2% agar (BD Sciences, 214040) to the media described above.

aTc was purchased from Cayman Chemicals (10009542) and prepared as a 4628.8 ng/mL (10 mM) stock in ethanol for long term storage at −20°C and diluted in water for experiments as necessary.

When constructing strains where $P_{7tet.1}$ replaces endogenous promoters, a PCR fragment containing $P_{7tet.1}$ and an antibiotic marker (either Nourseothricin (Werner BioAgents, clonNAT) or Hygromycin (ThermoFisher,10687010)) was transformed for homologous recombination directed replacement of the endogenous promoter. Cells were plated on YPD + antibiotic plates for selection. Whenever the promoter of an essential gene was being replaced, transformations were plated on multiple plates with YPD + antibiotic and 10/50/100/500 ng/mL aTc.

## Spotting assay

For spotting assays of cell growth and viability, cells were precultured in YPD media with 20 ng/mL aTc (except for WTC$_{846-K2}$::$IPL1$ strain which was precultured in 10 ng/mL aTc) and the necessary antibiotic to stationary phase, and diluted into YPD + antibiotic without aTc at a concentration of $0.8 \times 10^6$ cells/mL. Six hr later, cells were spun down and resuspended in YPD. Cells were spotted onto plates containing different media and aTc concentrations prepared as described above such that the most concentrated spot has $2.25 \times 10^6$ cells, and each column is a 1:10 dilution. Pictures were taken after 24 hr for the YPD and SD plates, and 42 hr for SD Proline, S Glycerol and YPE plates.

## Flow cytometry

Cells were diluted 1:200 from dense precultures and cultured to early exponential phase ($2–5 \times 10^6$ cells/mL) in 96 deep-well plates at 30°C before induction with aTc if necessary. For aTc dose responses, samples were taken at times indicated. For experiments where no dose response was necessary, cells were measured at least 4 hr after dilution of precultures, but always before stationary phase. Samples were diluted in PBS and measured using a LSRFortessa LSRII equipped with a high-throughput sampler. PMT voltages for the forward and side scatter measurements were set up such that the height of the signal was not saturated. Citrine fluorescence was quantified using a 488 nm excitation laser and a 530/30 nm emission filter. PMT voltage for this channel was set up such that the signal from $P_{TDH3}$ expressed Citrine did not saturate the measurement device, except for basal level measurements in *Figure 3B* and *Figure 3—figure supplement 3*, where PMT voltage for the Citrine channel was increased to maximum. Side scatter was measured using the 488 nm excitation laser and 488/10 nm emission filter.

## Western blots

Cells were grown to stationary phase with the indicated aTc concentration. 5 mL of cell culture was centrifuged and resuspended in 1 mL 70% ethanol. Fixed cells were again centrifuged, and resuspended in 200 uL Trupage LDS loading buffer (Merck, PCG3009) supplemented with 8M urea. Cells were broken using glass beads and a bead beater, and boiled at 95°C for 30 min. Proteins were separated using SDS-Page with Trupage precast 10% gels (Merck, PCG2009-10EA) and the associated commercial buffer, and transferred onto a nitrocellulose membrane (GE Healthcare Life Sciences, 10600008).

We used mouse monoclonal primary antibodies for detecting TetR (Takara, Clone 9G9), and Citrine (Merck, G6539), both diluted 1:2000 in Odyssey Blocking buffer (PBS) (LI-COR Biosciences) + 0.2% Tween 20. The secondary antibody was the near-infrared fluorescent IRDye 800CW Goat anti-Mouse IgG Secondary Antibody from Li-Cor (926–32210), diluted 1:5000 in the same manner. We used Chameleon Duo pre-stained Protein Ladder as our molecular weight marker (928-60000). We used the SNAP i.d. 2.0 system which uses vacuum to drive reagents through the membrane, and the Odyssey CLx (LI-COR) detector for imaging. Images were processed using the Fiji software to obtain black and white images with high contrast (*Schindelin et al., 2012*).

## Growth curves

Cells were precultured in YPD (with aTc in the case of strains where $WTC_{846}$ controlled essential genes) to stationary phase, then diluted into fresh media at a concentration of 50.000 cells per mL and induced with the necessary aTc concentrations, except for YP Ethanol and S Ethanol media where the concentration was 500,000 cells per mL. The Growth Profiler 960 (EnzyScreen) with 96-well plates and 250 µL volume per well, or Biolector (m2p-labs) with 48 well plates and 1 mL volume per well was used to measure growth curves. These are commercial devices that quantify culture density by detecting the light that is reflected back by the liquid culture.

## Arrest and release assay and DNA staining

$WTC_{846}$-*K3::CDC20* and the appropriate control strains were precultured in YPD (pH 4) with with the indicated aTc concentration to a concentration of $2x10^6$ cells/mL, then centrifuged and diluted 1:3 into YPD (pH 4) without aTc. We found that low pH (pH4) of the media was necessary for efficient mother-daughter separation upon completion of cytokinesis, potentially due to the low pH optimum of the chitinase CTS1 (*Hurtado-Guerrero and van Aalten, 2007*), which plays a role in separation. For the experiment presented in *Figure 5E*, to prevent the culture from becoming too dense, 25% of the media was filtered and returned to the culture after 4 hr of growth without aTc, which removed 1/4th of the cells. If release was performed, this was done after 8 hr of arrest by adding 600 ng/mL aTc to the culture. Samples were taken at indicated time points before, and every 5 min after aTc was added to the culture, and fixed with 70% ethanol. For the experiment presented in *Figure 5E*, to aid mother-daughter separation, the samples were sonicated for 1 min in a water bath before fixation.

Samples for DNA staining were digested with 5 mg/mL proteinase K for 50 min at 50°C, followed by 2 hr of RNase A (Applichem, A2760,0500) treatment at 30°C. Samples were stained for DNA content using SYTOX Green (Thermofisher, S7020) diluted 1:5000 in PBS, and were sonicated in a water bath for 25 s before flow cytometry. Fluorescence was detected using a 488 nm excitation laser and a 525/15 nm emission filter. The PMT voltage was set up such that the sample with the highest expected ploidy did not saturate the signal.

## Shutoff assay

Cells were grown to early exponential phase(~3 million cells/mL) in YPD at 30°C with shaking and induced with 600 ng/mL aTc. Two mL samples were taken at indicated time points. To remove excess aTc, cells were spun down for 20 s, supernatant was removed and cells were resuspended in YPD. This process was repeated three times. After the 3rd resuspension, the 2 mL sample was divided between two wells of a 96 deep-well plate. Cycloheximide was added to one of the wells at a final concentration of 70 µg/mL. The plate was continuously shaken at 30°C. Citrine fluorescence was measured every 30 min using flow cytometry as explained above.

## Data analysis

All analysis was performed using R (*R Development Core Team, 2013*), and the packages Bioconductor (*Ellis et al., 2009*), dplyr (*Wickham et al., 2018*), drc (*Ritz et al., 2015*), MASS (*Kafadar et al., 1999*), mixtools (*Benaglia et al., 2009*), and ggplot2 (*Ginestet, 2011*). All raw data that is not provided as source data here is available publicly at doi.org/10.3929/ethz-b-000488967.

Flow cytometry data was not gated except when necessary to remove debris. For aTc dose response experiments, median fluorescence of the entire population was used to fit a five-parameter dose response curve with the drm() command and the fplogistic formula $c + \frac{(d-c)}{1+\exp(b(log(x+1))^{p\_1}+\exp(log(x+1))^{p\_2})}$ from the drc package. Parameters p_1 and p_2 were fixed individually for each curve, the rest of the parameters were estimated by the drm command. Parameter values can be found in *Supplementary file 1* - Table S5. The cytometry cell volume proxy was always calculated as the magnitude of the vector of the FSC-W and SSC-H signals ($\sqrt{(FSC-W)^2 + (SSC-H)^2}$), since forward and side scatter signals provide information about cell volume and budding state. The forward scatter width and side scatter height were chosen because this combination (as opposed to other combinations involving FSC-H/SSC-W or area of the signals)

showed the most separation between measured signal peaks corresponding to spherical calibration beads of known diameter.

For single-reporter quantification of VIV, we calculated the residual standard deviation (RSD) of a linear model describing the relationship between the cytometry cell volume proxy and fluorescence of the population. To do this, the rlm() command from the MASS package was used to generate the linear model, and the residual standard deviation given by the same rlm() command was used as our measure of VIV. See Appendix 2 for a detailed explanation of the method.

Where shown, error bars for median fluorescence and the RSD were calculated using bootstrapping. The original set of data points was sampled with replacement and median fluorescence or RSD was calculated. 95% confidence intervals were calculated based on 1000 repetitions of this sampling process and plotted as error bars.

To generate a linear model describing the relationship between the volume proxy measurements done by flow cytometry and volume measurements by Coulter Counter, first the two data sets were sampled with replacement 5000 times. Then these samples were ordered by increasing volume proxy or volume and merged. The lm() command in the R package stats was used to fit the linear model. Sub-populations from Gates 4, 6, and 9 were not included in the fitting, as these medians were deemed suboptimal representations of the bimodal distributions. The resulting linear fit had a slope of 471 and an intercept of 62032.

## Acknowledgements

We thank Jörg Stelling for conceptual discussions, Hans Michael-Kaltenbach for discussions on the cell-to-cell variation measure, Kristina Elfström for initial construction of $P_{2tet}$ constructs, Gnügge for initial characterization of repressibility of operator placements around the TATA sequence by LacI, and Mattia Gollub, Justin Nodwell, Alan Davidson, Kohtaro Tanaka, and Alejandro Colman-Lerner for valuable discussions over the course of the work. This work was supported by the Swiss National Science Foundation as part of the Molecular Systems Engineering NCCR and by grant R21CA223901 from the NCI to RB.

## Additional information

### Funding

| Funder | Grant reference number | Author |
|---|---|---|
| Schweizerischer Nationalfonds zur Förderung der Wissenschaftlichen Forschung | NCCR Molecular Systems Engineering | Asli Azizoglu Fabian Rudolf |
| National Cancer Institute | R21CA223901 | Roger Brent |

The funders had no role in study design, data collection and interpretation, or the decision to submit the work for publication.

### Author contributions

Asli Azizoglu, Investigation, Visualization, Writing - original draft, Writing - review and editing; Roger Brent, Conceptualization, Supervision, Funding acquisition, Writing - review and editing; Fabian Rudolf, Conceptualization, Supervision, Writing - review and editing

### Author ORCIDs

Asli Azizoglu ![iD] https://orcid.org/0000-0002-2600-1322
Roger Brent ![iD] https://orcid.org/0000-0001-8398-3273

### Decision letter and Author response

Decision letter https://doi.org/10.7554/eLife.69549.sa1
Author response https://doi.org/10.7554/eLife.69549.sa2

# Additional files

## Supplementary files

- Supplementary file 1. Table S1 Strains used in this study. Kozak sequence is the last 15 bp before the start codon. Table S2 Plasmids used in this study. * indicates plasmids available through Addgene. These plasmids are sufficient to allow construction of WTC$_{846}$ strains carrying the cAR architecture without any further construction. They can also be modified to construct strains with genes controlled by the Simple Repression (SR) and Autorepression (AR) architectures presented in this manuscript. Construction of SR strains would require deletion of the negative feedback-controlled TetR from (P2365/2370/2371/2372/2374), and construction of AR strains would require deletion of the constitutively expressed TetR-Tup1 from the same plasmids. Table S3 Oligos used in this study to create strains where WTC$_{846}$ controls endogenous gene expression. See Appendix 5 for the protocol used for endogenous gene promoter replacement through homology directed repair. These oligos were used in conjunction with P2375 (NatMX) or P2350 (HygMX) to create the linear PCR fragment necessary for promoter replacement, or as colony PCR oligos to confirm correct promoter replacement. Table S4 Sequences used in this study. (*) indicates a shortened t_CYC1 used to avoid homology in plasmids where there are more than one t_CYC1 sequences. (**) indicates the linker sequence used between TetR-nls and the fusion partners MBP and Tup1. (***) indicates the linker sequence used between TetR-nls and the fusion partner GST. Table S5 Parameters used to fit 5-parameter sigmoid curves to experimental data. See Materials and methods for the 5-parameter log logistic forumula.

- Transparent reporting form

## Data availability

All relevant sequences are included in the supporting files for reproducibility. All raw flow cytometry data is publicly available at https://doi.org/10.3929/ethz-b-000488967. All other source data is included in the manuscript and supporting files.

The following dataset was generated:

| Author(s) | Year | Dataset title | Dataset URL | Database and Identifier |
|---|---|---|---|---|
| Azizoglu A | 2021 | A precisely adjustable, variation-suppressed eukaryotic transcriptional controller to enable genetic discovery | https://doi.org/10.3929/ethz-b-000488967 | ETH Zurich Research Collection, 10.3929/ethz-b-000488967 |

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

## Appendix 1

## Optimization of $tetO_1$ placement and endogenous transcription factor binding sites

When creating $P_{7tet.1}$, the final promoter used in $WTC_{846}$, we optimized the placement of the $tetO_1$ sequences, the sequences of the endogenous Gcr1 and Rap1-binding sites, the number of these sites present in the promoter, and the TATA sequence. Our goal was to increase maximum expression from the promoter, because $P_{TDH3}$ derivatives we had constructed with $tetO_1$ sites had shown reduced expression. For these optimizations we used as a starting promoter $P_{4tet}$. This promoter is a variant of $P_{5tet}$ (*Figure 1B*) from which the $tetO_1$ sequence immediately upstream of the TATA was removed. $P_{4tet}$ showed a higher basal expression level than $P_{5tet}$, allowing us to better observe subtle differences in basal expression (*Appendix 1—figure 1B*). We tested repression of $P_{4tet}$ and derivatives in the SR architecture. We constructed the derivatives as follows. We extended the Rap1 site and the upstream Gcr1 site by one base pair ($P_{4tet.1}$ and $P_{4tet.2}$), to account for the possibility that we initially truncated the endogenous binding sites *Metzger et al., 2015*, replaced the downstream Gcr1 binding site with the same extended Gcr1 site ($P_{4tet.3}$), since the upstream Gcr1 binding site was closer to the reported consensus sequence *Huie et al., 1992*, and tested alternative TATA sequences ($P_{4tet.4}$ and $P_{4tet.5}$) *Mogno et al., 2010*. Four of these optimizations resulted in increased maximum activity (*Appendix 1—figure 1B*): the Rap1 site extension ($P_{4tet.1}$), replacement of the downstream Gcr1 site ($P_{4tet.3}$), and the TATA sequence optimizations (with $P_{4tet.5}$ driving expression more strongly than $P_{4tet.4}$). $P_{4tet.2}$ with the extended Gcr1 site showed reduced maximum expression.

We therefore chose the following modifications: the single base pair extension of the upstream Rap1 site, the replacement of the downstream Gcr1 site with the original upstream Gcr1 sequence, and the TATA sequence TATAAATA. We implemented these modifications to $P_{5tet}$ to generate $P_{5tet.1}$. By the assays described in the main text for testing the other promoter derivatives, compared to $P_{5tet}$, the new promoter $P_{5tet.1}$ (Y2659) showed 95% of maximum expression driven by $P_{TDH3}$, and increased repression (15 fold vs. 12 fold) with only a slight increase in basal activity when fully repressed (Y2656, *Figure 1B* and *Appendix 1—figure 1C*). In order to increase maximum expression even further, we took advantage of previous work showing that increasing the number of transcription factor binding sites in a promoter could increase its strength *Ottoz et al., 2014*. We investigated whether adding additional Rap1 and Gcr1 sites to $P_{5tet.1}$ would increase promoter activity. We created $P_{7tet}$ by duplicating the Rap1 site, and $P_{7tet.1}$ by duplicating both the Rap1 and one of the two Gcr1 sites in $P_{5tet.1}$, while keeping the same $tetO_1$ placements at these duplicated sites (*Appendix 1—figure 1C*). $P_{7tet.1}$ had a higher maximum activity (116% vs 99% of $P_{TDH3}$ activity) and fold repression than $P_{7tet}$ (20-fold vs 18-fold), with only minimal increase in absolute repressed activity (4.3-fold vs fourfold above autofluorescence). We therefore chose $P_{7tet.1}$ as the promoter for further use.

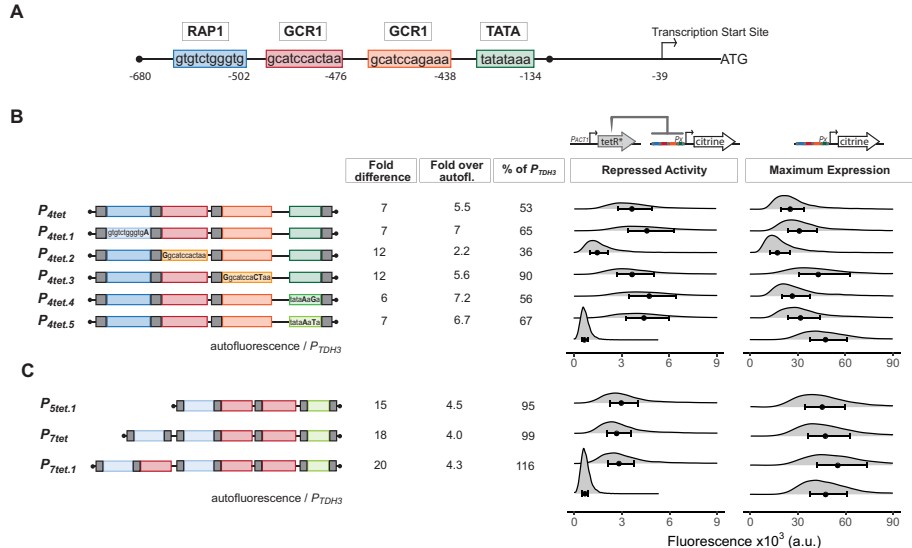

**Appendix 1—figure 1.** Optimization of $tetO_1$ placements and endogenous transcription factor binding sites to increase maximum activity of a TetR-repressible derivative of $P_{TDH3}$. (**A**) Diagram of $P_{TDH3}$ shows the nucleotide positions of the binding sites for the endogenous transcription factors Rap1 and Gcr1, the TATA-sequence, and the transcription start site relative to the TDH3 start codon. Gcr1 binding sites were found in reference *Yagi et al., 1994* and confirmed in reference *Kuroda et al., 1994*. (**B**) Repression and maximum activity of the $P_{TDH3}$ derivatives tested for optimization. Diagrams above the plots display the genetic elements of strains used (Y2565, 2575,2598,2647,2599,2648,2601,2649,2602,2650,2603,2656,70,2683). Left diagram depicts strains used to test repressed activity, right diagram maximum activity. Px denotes any tetR repressible promoter. The * in TetR indicates a SV40 Nuclear Localization Sequence. In all strains, the $P_{TDH3}$ derivative promoters diagrammed on the left directed the synthesis of Citrine integrated into the *LEU2* locus. Grey boxes inside the diagrams denote $tetO_1$ TetR-binding sites. For measurement of repressed activity, otherwise-isogenic strains carried a $P_{ACT1}$-TetR construct integrated in the *HIS3* locus. Citrine fluorescent signal was detected by flow cytometry. For the measurements, 'fold difference' measures the median of the maximum activity signal divided by the median of the repressed activity. 'Fold over autofluorescence' refers to median repressed activity signal divided by the median autofluorescence background signal. Maximum promoter activity is quantified as median fluorescence signal expressed as percentage of signal from otherwise-isogenic $P_{TDH3}$-Citrine strain. For the plots, x axis shows intensity of fluorescence signal. Plots are density distributions of the whole population, such that the area under the curve equals one and the y axis indicates the proportion of cells at each fluorescence value. The circles inside each density plot show the median and the upper and lower bounds of the bar show the first and third quartiles of the distribution. (**C**) Repression and maximum activity of optimized $P_{5tet}$ derivatives. Diagrams and plots as in (**B**). These promoter variants contained additional binding sites for Rap1 and Gcr1 selected for higher activity, as well as an alternative TATA sequence as described.

The online version of this article includes the following source data is available for figure 1:

**Appendix 1—figure 1—source data 1.** Numerical data for *Appendix 1—figure 1*.

## Appendix 2

## Single reporter quantification of CCV in fluorescent protein expression by flow cytometry using the Volume-independent variation measure

In yeast and *C. elegans*, comparison of signals from strains with different combinations of different reporter genes allows the different contributions to variation to be independently quantified (*Colman-Lerner et al., 2005*; *Mendenhall et al., 2015*). One of these contributions, individual differences in general ability to express genes into proteins, contributes to phenotypic variation in genetic penetrance and expressivity (*Burnaevskiy et al., 2019*). Quantification of this and other sources of variation benefits from ability to measure output of single cells over time (*Colman-Lerner et al., 2005*) and, in flow cytometry, requires measurement of outputs of multiple reporters (*Pesce et al., 2018*). Here, however, we were interested in the overall variability rather than specific sources of variability. Therefore, we only had a single reporter protein (Citrine). However single reporter studies have a major, confounding contribution to measured variation in gene expression that multi-reporter studies don't: Fluorescent proteins in yeast are degraded very slowly unless they have degradation tags attached (*Gordon et al., 2007*) and therefore, if constitutively expressed, their abundance increases over time (*Cookson et al., 2010*). Thus, in cycling populations of budding yeast that continually express fluorescent proteins, a major source of cell-to-cell variation in fluorescent signal is that small, new-born cells have not had time to accumulate much fluorescent protein, while larger cells have. This source of variability normally affects all reporter proteins in the cell in a similar fashion, and therefore does not require correcting in multi-reporter studies. On the other hand in single reporter studies with flow cytometry in yeast, as for higher cells, this volume related variation in fluorescent protein expression is generally corrected for by gating; that is filtering the data to select only a narrow subset of cells with similar forward and side scatter, and thus volume, which increases with cell cycle progression. Such gating disregards data from the majority of the cells whose values fall outside the gated range. Here, in order to avoid discarding data, we established a single-reporter measure of cell-to-cell variation that corrects for variation due to fluorescent protein accumulation without gating.

We first established that forward and side scatter signals can be used to distinguish smaller cells from larger ones. We sorted cells on a BD FACS Aria III flow cytometer. We set different gates on the FSC and SSC signals (shown in *Appendix 2—figure 1*) to collect 10 sorted sub-populations, each containing about 100,000 exponentially growing Y2683 cells. We then immediately measured (a) FSC and SSC from the collected subpopulations on a different instrument, the LSRII Fortessa LSR used for the flow cytometric measurements in this work, and (b) volume in fL with a Coulter Counter (*Appendix 2—figure 2*). The raw data acquired by the two methods can be seen in *Appendix 2—figure 2A,B*. For the flow cytometry data, we used the width of the FSC and height of the SSC to calculate a volume proxy using the formula $\sqrt{(FSC-W)^2 + (SSC-H)^2}$ as explained in Materials and methods. *Figure 2C* shows a linear relationship between the medians of the sub-populations as measured by the two methods, that is, that the flow cytometric measurement is a proxy for volume, and that two volume measurements qualitatively agree. The three sub-populations (4, 6, and 9) where a slight deviation from the linear relationship is observed are all bimodally distributed, meaning the population is a mixture of large and small cells and the median is not a good representation of this sub-population. Overall, for the cell-to-cell variation calculations outlined below, this relative relationship is enough to distinguish new-born, smaller cells from larger cells that have had time to accumulate fluorescent protein.

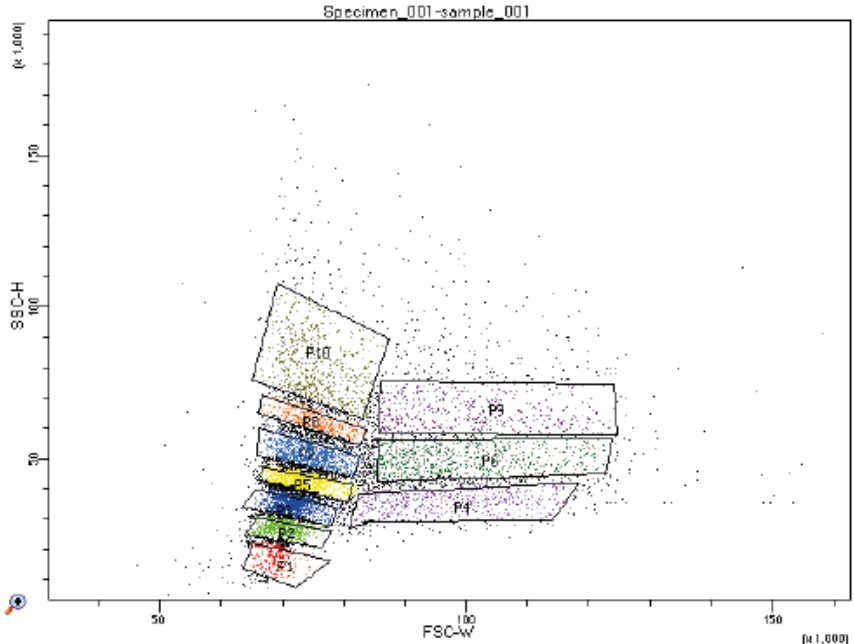

**Appendix 2—figure 1.** Sub-populations collected by FACS for validating the cell volume proxy measure. Strain Y2683 was grown to exponential phase in YPD and was run through the sorter at a concentration of 2 million cells per mL. 10 separate gates were set on the FSC-W and SSC-H signals for collecting sub-populations as depicted in the figure.

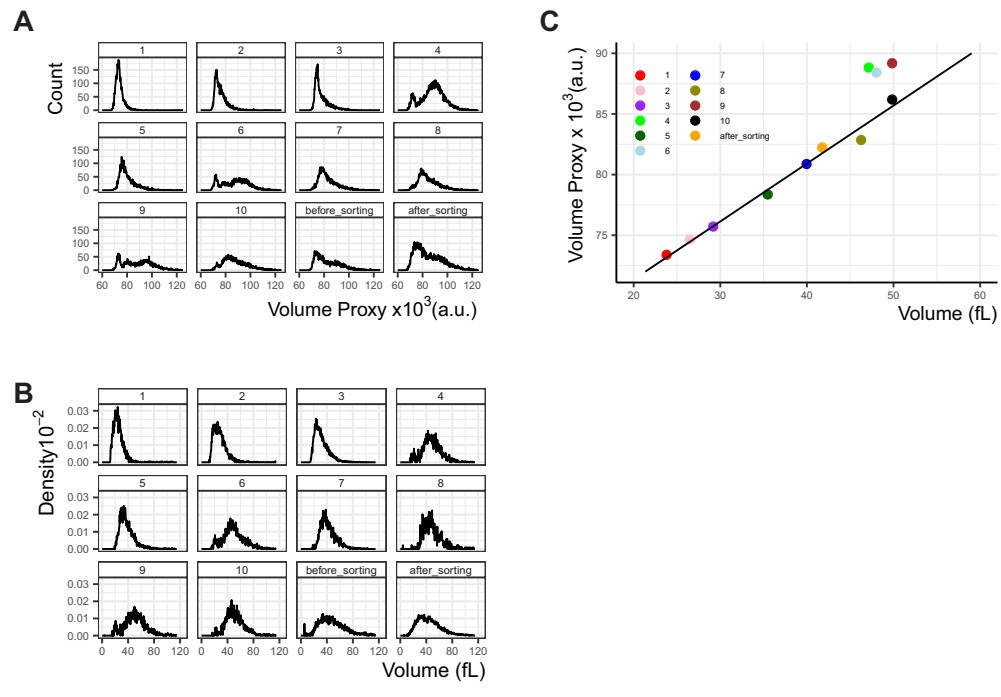

**Appendix 2—figure 2.** Comparison of the flow-cytometry-based cell volume proxy to cell volume measured by Coulter counter. 10 sub-populations were collected from an exponentially growing culture used as constitutive Citrine expression control in other experiments (Y2683), using FACS as

*Appendix 2—figure 2 continued on next page*

*Appendix 2—figure 2 continued*

described in Figure *Appendix 2—figure 1*. Each of the sub-populations were (**A**) measured in the LSRII Fortessa LSR flow cytometer used for all other experiments and the proxy for cell volume calculated and (**B**) measured in a Coulter counter. The sub-population numbers are indicated above the plots and correspond to the gates seen in Figure *Appendix 2—figure 1*. The culture was kept on ice throughout the sorting process, and the full population was measured twice, once before the sorting process began (before_sorting) and once after (after_sorting) to ensure the volume distribution in the population did not change over the course of the experiment. No significant difference was observed. (**C**) The median volume of each sub-population was plotted, as measured by flow cytometry (y axis) or Coulter counter (x axis). The linear fit was generated as explained in Materials and methods ($R^2$=0.98, p=7.37x10$^{-7}$), without taking into account gates 4, 6, and 9 where the distribution is bimodal and the median is not a good descriptor of the population. (**A**) displays cell counts per volume proxy, and (**B**) displays density plots where the area under the curve is 1.

The online version of this article includes the following source data is available for figure 2:

**Appendix 2—figure 2—source data 1.** Numerical data for *Appendix 2—figure 2C*.

We then used this information to calculate the CCV in fluorescent protein expression that could not be attributed to differences in cell volume/cell cycle progression, and called this method Volume Independent Variation (VIV). We began with Y2683 cells, which express Citrine from wild type $P_{TDH3}$. We then plotted the volume proxy vs. the fluorescence signal observed in the entire population measured by flow cytometry (*Figure 2—figure supplement 2*). Then we performed a robust linear fit on the cell volume proxy vs. the Citrine fluorescence signal. This linear model allowed us to correct for the differences in cell volume and calculate the RSD of the fit as explained in the Materials and methods. This RSD value quantifies the variation in the population that is not due to differences in cell volume between the cells. While use of this measure is in principle akin to measuring variation in expression of fluorescent proteins using a very narrow gate on the measured FSC vs SSC signals of the population, it avoids the need to discard data. Moreover, it could also be used when comparing populations with different cell volume distributions.

## Appendix 3

### Increasing the mass and nuclear concentration of TetR in order to abolish basal expression from $P_{7tet.1}$

We reasoned that basal expression from $P_{7tet.1}$ in the WTC$_{846}$ architecture might arise because (a) the nuclear concentration of TetR might be too low for all of the $tetO_1$ TetR binding sites in $P_{7tet.1}$ to be occupied at all times, and/or (b), that TetR derivatives might fully occupy all of the operators and yet not repress completely. We tested the first idea by increasing the nuclear concentration of TetR proteins by expressing derivatives that contained a second SV40 Nuclear Localization Sequence. We tested the second idea by fusing TetR to other protein moieties that might aid repression. Specifically, we added to TetR portions of prokaryotic proteins that we could presume to be inert, hoping that these bulkier TetR derivatives might repress more strongly, for example by better sterically interfering with the binding of transcription factors, or with contacts between Gcr1 and Rap1 at the UAS and the transcription apparatus at the core promoter. We tested the efficacy of these new molecules by expressing them from $P_{ACT1}$ in the SR architecture (Y2681, Y2664, Y2665, Y2666, and Y2667, *Appendix 3—figure 1*). For smaller repressors, addition of a second NLS decreased uninduced expression whereas for larger repressors it did not. The strain carrying the TetR-nls-MBP (Maltose Binding Protein, the *E. coli* malE gene product), showed the most repression, but still exhibited uninduced expression signal of 2.2-fold above autofluorescence background.

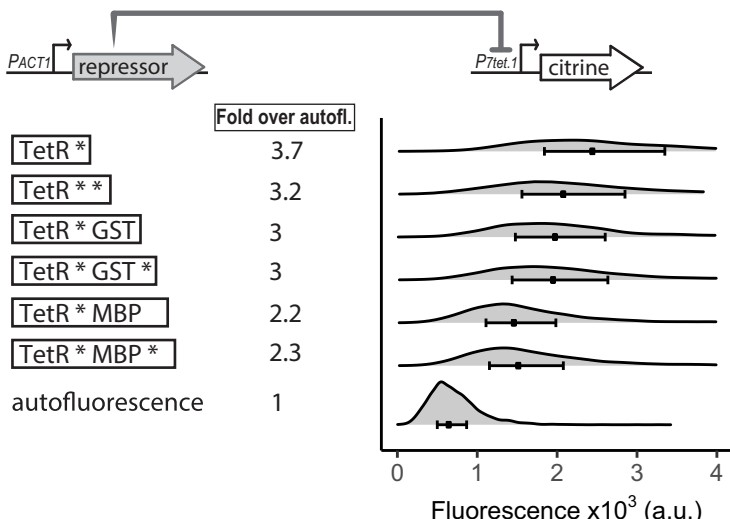

**Appendix 3—figure 1.** Effect of increased nuclear concentration and size of TetR on repression. The top diagram indicates the genetic elements of the SR architecture used to test the ability of various TetR derivatives to abolish basal activity of $P_{7tet.1}$. Diagrams to the left of the plots show the different repressors used. Each * indicates one SV40 Nuclear Localization Sequence. GST refers to Glutathione S-transferase, and MBP to Maltose Binding Protein, both of *E. coli*. Citrine fluorescence from $P_{7tet.1}$ repressed by the repressors indicated was measured using flow cytometry (Y2657,2681,2664,2665,2666,2667,70). Plots are density distributions of the whole population, such that the area under the curve equals one and the y axis indicates the proportion of cells at each fluorescence value. The circles inside each density plot show the median and the upper and lower bounds of the bar correspond to the first and third quartiles of the distribution. Numbers to the left of the plot indicate fold expression over autofluorescence, that is, the median of the Citrine fluorescence detected divided by the median of the autofluorescence signal. Although increased

*Appendix 3—figure 1 continued on next page*

*Appendix 3—figure 1 continued*

nuclear concentration and size of TetR increase repression efficiency, these strategies are not enough to fully abolish basal expression from $P_{7tet.1}$.

The online version of this article includes the following source data is available for figure 1:

**Appendix 3—figure 1—source data 1.** Numerical data for *Appendix 3—figure 1*.

## Appendix 4

### Time to steady state depends on the stability of the controlled protein

In the experiments in *Figure 4*, WTC$_{846}$-controlled Citrine takes around 7 hr to reach steady state concentration. Here we present a simple ODE model to demonstrate that the time to steady state will change based on the stability of the controlled protein. In the model (*Equation 1*) the protein of interest is produced at a constant rate $a$ and lost with a lumped linear rate (dilution + degradation) $d$. Analytical solution (*Equation 2*) of this model shows that the only constant that affects the time variable is the degradation + dilution rate. Dependence on this variable is also evident in simulations based on this model (*Appendix 4—figure 1*). The smaller $d$ is, (i.e. the more stable the protein is), the longer it takes to reach steady state. On the other hand, changes in the production rate $a$ have no effect on time to steady state, although both rates affect the maximum level. Citrine is a remarkably stable protein (see *Figure 4—figure supplement 8*), but recent data suggests that most (somewhere between 50–85% depending on the data set) of the yeast proteome is just as stable *Wiechecki et al., 2018*. Therefore most other WTC$_{846}$-controlled endogenous proteins will likely have a time to steady state around 7 hr, except those with a shorter half-life which will exhibit a shorter time to steady state.

$$\frac{d[P]}{dt} = a - d \cdot P \tag{1}$$

$$[P] = \frac{a}{d}\left(1 - e^{-dt}\right) \tag{2}$$

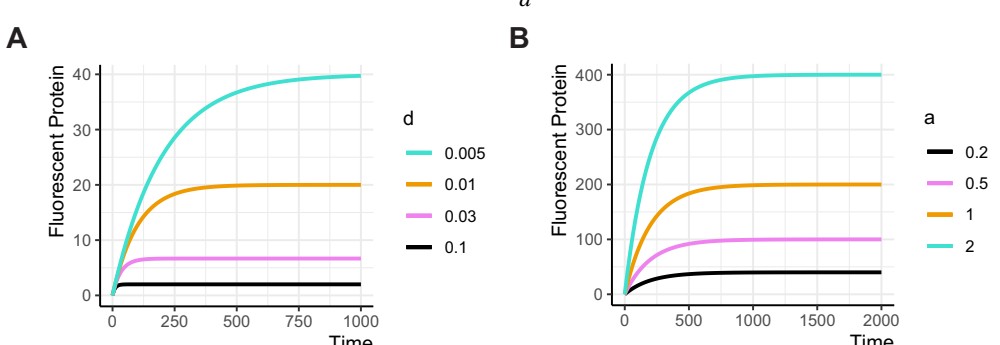

**Appendix 4—figure 1.** Simulations showing the relationship between production and degradation rates and time to steady state. The ODE model presented in Appendix 4 was simulated with (**A**) constant production rate $a$ = 0.2, and a varying degradation rate $d$, or (**B**) with constant degradation rate $d$ = 0.005 and a varying production rate. Both parameters affect the maximum level of Protein produced, but only the degradation rate determines the time required to reach the steady state expression. The lower the degradation rate (i.e. the more stable the protein is), the longer it takes to reach steady state.

## Appendix 5

### Protocol for WTC$_{846}$ strain generation

The WTC$_{846}$ is a two unit transcriptional control system for *S. cerevisiae* (*Appendix 5—figure 1A*). An inducible promoter ($P_{7tet.1}$) is placed in front of the Gene of Interest (GOI). The promoter is based on an engineered version of the strong constitutive promoter of *TDH3*. It was made repressible by placing TetR-binding sites next to the binding sites for the transcriptional machinery. As a result, binding of the TetR protein can prevent binding of the endogenous proteins which normally drive transcription. The repressors TetR and TetR-Tup1 are found on one integrative repressor plasmid (*Appendix 5—figure 2A*). TetR is expressed under the above described promoter ($P_{7tet.1}$) creating an autorepression loop. TetR-nls-Tup1 abolishes the basal activity of $P_{7tet.1}$ and is expressed under the control of the weak, constitutive *RNR2* promoter.

To create a functional system, we advise to first integrate the repressor plasmid. The $P_{7tet.1}$ can then be placed in front of any gene in the genome using PCR tagging *Janke et al., 2004*. The tagging plasmid (based on *Janke et al., 2004*) is used as a template (*Appendix 5—figure 3*). We provide two versions of the $P_{7tet.1}$ followed by a flag tag followed by a linker composed of eight glycine residues; either cloned in a plasmid providing a HygR marker (P2350), or a NAT marker (P2375). For PCR-based tagging, the 5' and 3' ends of the PCR fragment need to be complementary to a sequence upstream of the GOI and to the beginning of the GOI, respectively. This is ensured by using primers with tails complementary to these regions. We tested the plasmid for use with and without the flag tag.

**A**

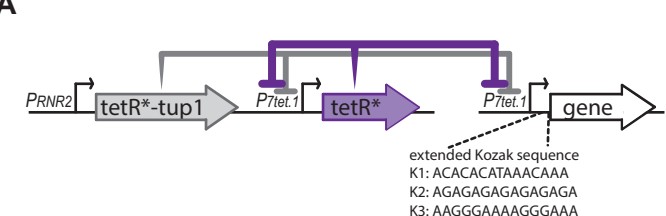

extended Kozak sequence
K1: ACACACATAAACAAA
K2: AGAGAGAGAGAGAGA
K3: AAGGGAAAAGGGAAA

**B**

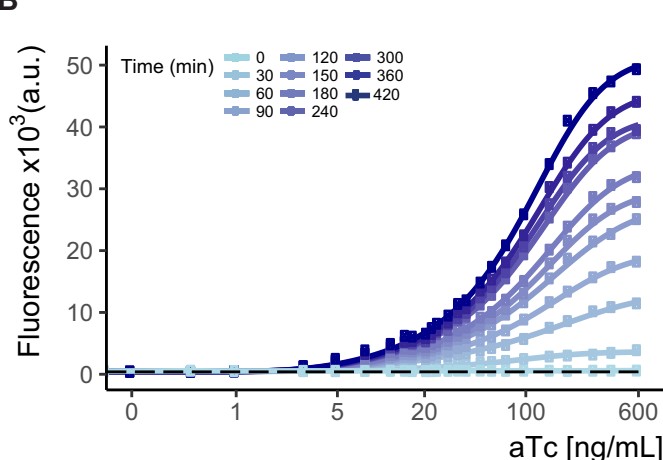

**Appendix 5—figure 1.** The configuration of and an example of gene expression control by WTC$_{846}$. (**A**) Genetic elements of the WTC$_{846}$ controller. On the integrative plasmid, TetR is driven by the $P_{7tet.1}$, TetR-nls-Tup1 is driven by the *RNR2* promoter. The promoter of the gene of interest is replaced with $P_{7tet.1}$ in the genome. (**B**) WTC$_{846}$ controlled Citrine expression. Flow cytometry measurements from a strain where WTC$_{846}$ regulates expression of Citrine. aTc was added to exponentially growing cells, and samples were taken every 30 min for flow cytometry analysis. Circles represent the median of the fluorescence signal, lines were fitted. The dashed line indicates autofluorescence control, that is, the parent strain without any Citrine integrated.

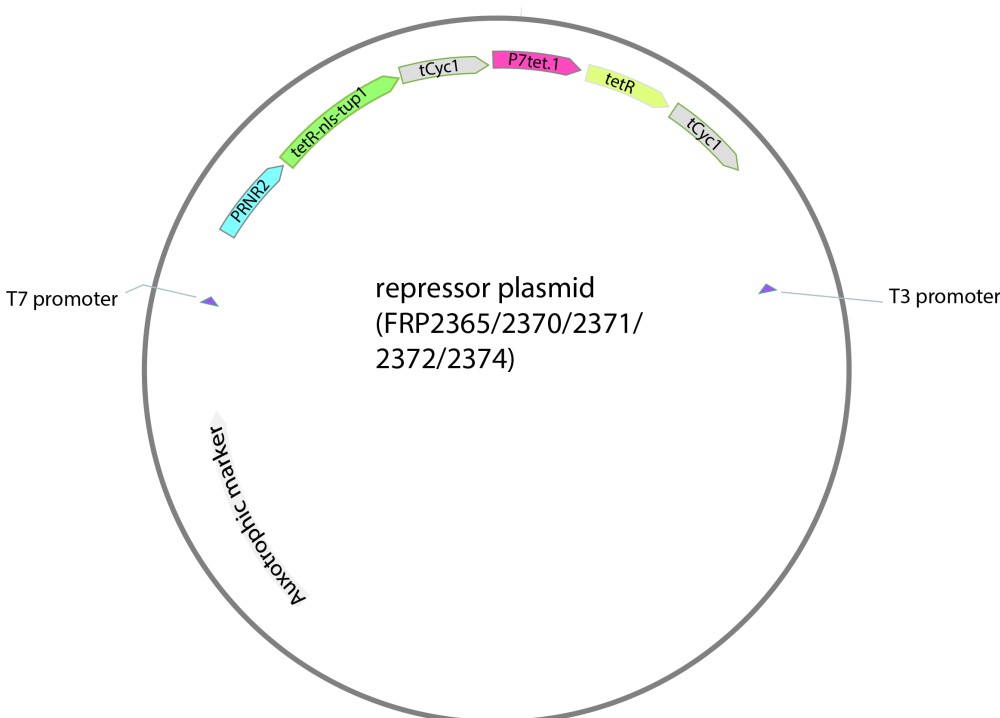

**Appendix 5—figure 2.** Map of the repressor plasmid. Auxotrophic marker is different depending on the plasmid backbone.

Induction of the tagged gene can then be controlled by aTc, a small molecule that causes TetR to dissociate from its binding sites on $P_{7tet.1}$. An example is seen in *Appendix 5—figure 1B*, where Citrine expression was controlled across a large expression range using aTc. Tetracycline or Doxycycline can also be used, although they will likely require different concentrations compared to aTc.

The basal activity of $P_{7tet.1}$ can be controlled by the Kozak sequence (last 15 bp before the start codon of the gene of interest). The provided sequence in the P2350 and P2375 plasmids shows no detectable basal Citrine expression. However, even a small basal expression level can become an issue if the GOI encodes a protein that is required in very small numbers. We encountered this problem with Tor2, Cdc28 and similarly low abundance, stable proteins. In this case, changing the translation efficiency by modifying the Kozak sequence allowed us to abolish all basal expression. The protocol below also explains how to achieve this.

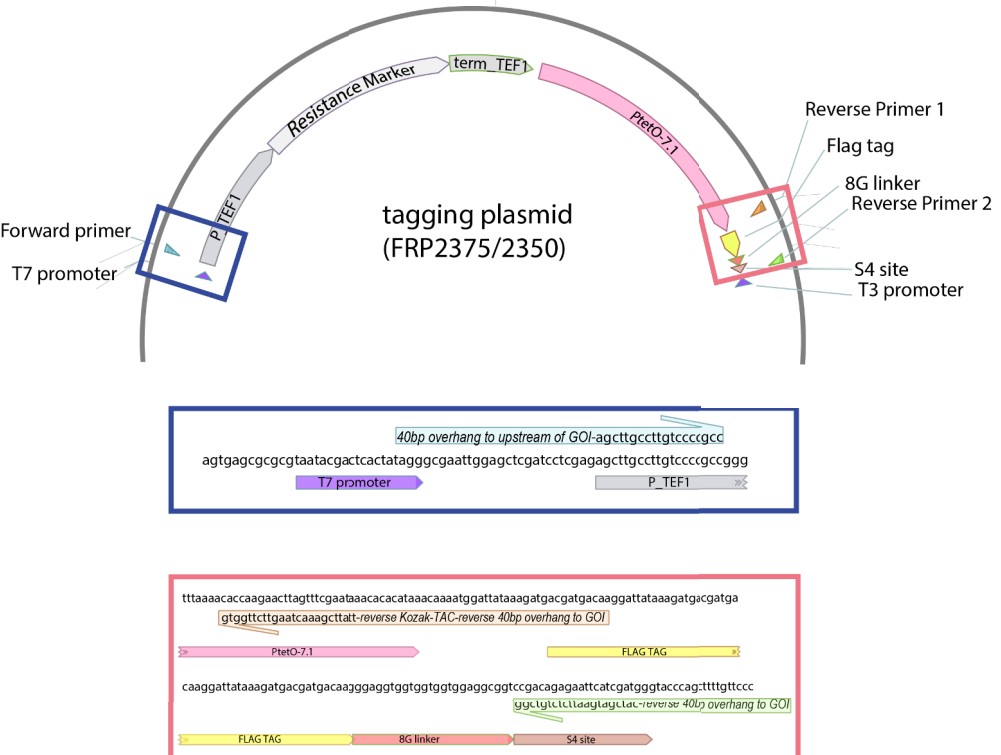

**Appendix 5—figure 3.** Map of the tagging plasmid. Resistance marker is NAT or HygR depending on the plasmid number. The colored boxes zoom in to the marked regions to demonstrate how the primers anneal to the plasmid.

## Tagging protocol

1. Transform the repressor plasmid in a strain that has the correct auxotrophic marker deletion. The plasmid should be linearized using AscI digestion for integrative transformation *Gnügge et al., 2016*. The repressor plasmids are given in *Appendix 5—table 1*:

2. Design primers to create the tagging fragment from the tagging plasmid (P2350 or P2375).

   ○ *Forward primer:* Use the sequence agcttgccttgtccccgcc as the annealing part of the forward primer. Select 40 base pairs anywhere upstream of the GOI, and use this sequence as the 5' tail of your forward primer. Remember that the region between these 40 base pairs and the start codon of the gene will be deleted during the transformation. You can thus remove the entire natural promoter of the gene, but this is not mandatory.

   ○ *Reverse primer option 1 - without flag tag:* Take tttattcgaaactaagttcttggtg as the annealing portion of your reverse primer, which will anneal to the sequence caccaagaacttagtttcgaataaa on the plasmid. Then use the reverse complement of the first 40 base pairs (including ATG) of the GOI, followed by the reverse complement of the desired Kozak sequence as your 5' tail, such that the primer reads: 5'-reverse GOI sequence-CAT-reverse Kozak sequence-annealing portion-3'.

   ○ *Kozak sequence to modulate expression:* The Kozak sequences that we have tested, in decreasing order of translation efficiency are (reverse complement is given in parentheses):
      • ACACACATAAACAAA (TTTGTTTATGTGTGT)
      • AGAGAGAGAGAGAGA (TCTCTCTCTCTCTCT)
      • AAGGGAAAAGGGAAA (TTTCCCTTTTCCCTT)

   ○ *Reverse primer option 2 - including flag tag:* If you would like to include the flag tag at the start of the gene, use the sequence catcgatgaattctctgtcgg as the annealing portion of your reverse primer, which will anneal to the standard S4 primer binding site on the plasmid (ccgacagagaattcatcgatg) . In this case the Kozak sequence cannot be altered, and the

one already on the tagging plasmid has to be used (this is the first one in the list above). Use the reverse complement of the first 40 bases (after ATG) of the gene as the 5' tail of your reverse primer such that the primer reads: 5'-reverse GOI sequence-annealing portion-3'.

3. Perform the tagging PCR to generate the tagging fragment using the primers designed in the previous step. If your standard PCR protocol fails (occasionally happens due to the long tails on the oligos), use the PCR protocol detailed below (adapted from *Janke et al., 2004*):Reaction Setup (200μL): 20μL Taq/Vent Buffer*, 35μL 2mM dNTPs, 2μL tagging plasmid, 0.5μL 100μM forward primer, 0.5μL 100μM reverse primer ,0.8μL Taq polymerase, 0.4μL Vent polymerase, 134.4μL ddH$_2$O. (*Buffer composition: 500mM Tris/HCl(pH=9.0) 22.5mM MgCl$_2$ 160mM NH$_4$SO$_4$). PCR program: (a) 95°C 5min (b) 95°C 1min (c) T$_a$ 30sec (d) 68°C 1min per kb (e) 95°C 1min (f) T$_a$ 30sec (g) 68°C 1min/kb + 20sec per cycle (h) 9°C hold. Repeat steps (b-d) 10x, (e-g) 20x.

4. Gel isolate and transform the tagging fragment into the strain created in step 1. Select on solid medium with the appropriate antibiotic and aTc. If the GOI is an essential gene, the transformation efficiency will be low. In order to increase transformation efficiency, pre-culture, recovery media for the cells and selection plate should all contain aTc. Note that the required aTc concentrations are around five times higher in solid media than in liquid media to achieve the same expression level.

5. Correct integration can be confirmed using colony PCR. Use sequence cagttcgagtttatcattatcaatactg as the forward primer (binds at the start of $P_{7tet.1}$), and a reverse primer that anneals within the GOI. The fragment length will depend on where in the GOI the reverse primer anneals. (This forward primer will work for all cases except when the *TDH3* promoter is being replaced. Since $P_{7tet.1}$ is based on the *TDH3* promoter, this primer will anneal to the promoter whether or not the replacement was successful.) Integration efficiency is low when tagging essential genes (about 10% of colonies screened), but a positive PCR result generally is enough to indicate correct integration. However it is best to isolate the PCR fragment and sequence the entire promoter to confirm correct integration.

**Appendix 5—table 1.** Repressor plasmids.

| P number | Marker | Backbone pRG number from Gnügge et al. (2016) |
| --- | --- | --- |
| P2365 | URA3 | pRG206 |
| P2370 | LEU2MX | pRG205MX |
| P2371 | HIS3MX | pRG203MX |
| P2372 | LYS2 | pRG207 |
| P2374 | MET15 | pRG201 |

