## [Decision Letter]

[Editors' note: this paper was reviewed by Review Commons.]

---

## [Author Response]

Summary review comments and responses

Both reviewers were obviously expert and their comments positive and constructive. Taking them together, the major comments we received on our manuscript fall into three different categories.

1. Need for clarification, either in terms of wording (Reviewer 1, #1) or more detailed explanation of the design process (R1, #2)

2. Requests for us to further elaborate on our results and their implications (R1, #3,4,6,7,9,10,11)

3. Requests for additional characterization of the system (R1, #5,8). In this category there was also one minor comment from Reviewer 2 that requested additional controls.

Actions we have taken.

We have addressed all major and minor comments and performed additional experiments where necessary.

1. We adjusted the wording as suggested and added text to clarify the design process.

2. We responded to each comment in this category in detail.

a. R1#3, we added text and a simple computational model for clarification.

b. R1#4,7,9,10,11 we added text to further elaborate on the points raised.

c. R1#6, we added text and performed an additional experiment to further elaborate on our results.

3. We performed all the requested additional characterization and control experiments.

a. R1#5, we performed two additional experiments to further characterize the shut-off speed of the system.

b. R1#8, as requested, we characterized two other transcriptional control systems and compared them to our system in terms of cell-to-cell variation.

c. R2#m1, we performed the control experiment requested.

In addition, we addressed all minor comments as suggested by the reviewers.

Detailed comments and responsesReviewer #1 (Evidence, reproducibility and clarity (Required)):SummaryAzizoglu et al. describe a novel system for highly regulated, conditional expression of any gene in budding yeast, with the goal of facilitating quantitative functional genetic analysis. The authors aimed to meet 6 criteria in their system: (1) operation in all growth media; (2) regulation by an exogenous small molecule with minimal effect on cell function; (3) zero basal expression to obtain null phenotypes; (4) large range of precisely adjustable expression; (5) very high maximal expression; and (6) low cell-to-cell variability at any output level. The data shown indicate that they have more or less achieved these aims and that the system they describe is an advance on current methods. The logic behind the development of the system is very clearly presented, as well as its relationship to previous work. In addition, the authors place their work in a very broad historical perspective, which is unusual but much appreciated by this reviewer. Finally, they complement their description of the system (which employs a fluorescence read-out) with several case studies on a variety of different yeast genes that demonstrate its utility and highlights its advantages relative to the current state-of-the-art methods. This study thus provides an important new tool for quantitative analysis of the effect of expression level on phenotype that should be of wide interest in the yeast genetics community as a discovery tool.

We appreciate this generous summary.

We note that the approach to setpoint gene regulation is general beyond yeast. We expect it to be applied to vertebrate systems and we expect it to be applied to plants. We also note that, although its primary application is genetic discovery, it is finding use in some biotechnology applications, for example for tuning the expression of genes in multistep biochemical pathways to find a level that results in maximum synthesis of an end product.

Primary comments and suggestions1. The phrase "Expression of genes conditionally into phenotype…", used in the Abstract, and in other guises elsewhere in the text, is awkward and confusing. A more comprehensible alternative could be: "The relationship between gene expression level and phenotype remains central to biological research. Current methods enable either on/off or imprecisely controlled graded expression…"

We thank the reviewer. In response, we changed the wording.

New abstract now reads "Conditional expression of genes and observation of phenotype remain central to biological discovery".

The variant of the awkward phrase occurs in one other place, in the first sentence of the introduction. In response to the reviewer, we changed the revised text to read

“… means to express genes conditionally to permit observation of the phenotype have remained.…”

Old text read:

“… means to express genes conditionally into phenotype have remained.…”

2. It seems to me that the AR strain (Figure 2A, and Supplementary Figure 4) displayed less cellto-cell variation than the cAR variant. Why was this strain not pursued further as an alternative for cases in which this feature might be more desirable?

We thank the reviewer for this question. We should have been clearer.

Yes, the AR strain displays lower cell-to-cell variation compared to the cAR strain in Figure 2A, especially at lower doses of aTc. But this was not true for the final cAR constructions that comprise WTC_846_ (compare Figure 4 and the AR strain in Figure 2). When comparing the AR strain in Figure 2 and the final WTC_846_ strain in Figure 4, at lowest expression levels, cell-to-cell variation is almost the same.

There are two differences between the cAR strain presented in Figure

2 and the final WTC_846_ strain: the former has TetR instead of TetRTup1 as the constitutively expressed element of the architecture, and more importantly, the promoter for this constitutively expressed repressor in the original cAR strain (P_ACT1_) is much stronger than in the final WTC_846_ strain (P_RNR2_). As a consequence, the starting cAR strain constitutively expressed a much larger amount of repressor, which was not subject to the cell-to-cell variation reducing effects of the negative feedback loop. The lower constitutive expression of repressor in the WTC_846_ strain means that a greater proportion of the cellular complement of repressor is autorepressed, and variation is suppressed.

The answer to why we pursued the cAR architecture after the experiments in Figure 2 lies in the initial criteria we had set for our ideal transcriptional controller. These are enumerated in the Introduction section. The two relevant criteria read: "(3) manifest no basal expression of the controlled gene in absence of inducer, allowing generation of null phenotypes” and "(6) exhibit low cell-to-cell variability at any set output…” In the case of our AR strain, we reasoned we could never fulfil criterion (3), as long as both the repressor and the controlled gene were expressed from the same TetR repressible promoter. If we wished the gene of interest to have no basal expression, then by definition TetR should have no basal expression, which would then mean there is no TetR to repress the promoters.

The cAR strain therefore embodies a trade-off between being able to supress basal expression and allowing higher cell-to-cell variation within a small portion of the dynamic range of the controller. We went with this trade-off because we judged that the ability to zero out basal expression was important for generating “reversible knockout” phenotypes, and perhaps in some synthetic biology applications where complete absence of a protein might be required for proper function of an engineered synthetic genetic circuit or to keep a kill switch off. Additionally, we went with the trade off because the means by which we would achieve it, the "zeroing repressor" concept, would be applicable to other WTC_846_-like controlled systems. As a backup, we made parts available via Addgene that enable construction of strains with the other architecture, AR, by one step cloning.

In response to the reviewer comments, we made the following changes.

Last paragraph of the Results section titled “Complex autorepressing (cAR) controller architecture expands the input dynamic range and reduces cell-to-cell variation”:

Previously the text read:

“Compared with cells bearing the SR architecture, otherwise-isogenic cells bearing the AR architecture showed increased basal expression (6.3 vs. 4.1fold over autouorescence background). […] We therefore picked this cAR architecture for our controller.”

The text in the Results section now reads:

“Compared with cells bearing the SR architecture, otherwise-isogenic cells bearing the AR architecture showed increased basal expression (6.3 vs. 4.1fold over autofluorescence background). […] We therefore picked this cAR architecture for our controller.”

We also made the following addition to the Discussion:

“Cell-cell variation in WTC_846_-driven expression is highest at low aTc levels, because control in this regime depends mostly on the higher variability Simple Repression by TetR-Tup1 expressed from P_RNR2_. […] This autorepressing circuitry operationally defines WTC_846_ as an ”expression clamp”, a device for adjusting and setting gene expression at desired levels and maintaining it with low cell-to-cell variation, and so allowing expressed protein dosage in individual cells to closely track the population average.”

We also made the following addition to the legend in the Plasmids table (Supplementary Table 2), to make it clear that we have provided information and materials sufficient to make strains with the other architectures used in this study.

“Plasmids used in this study. * indicates plasmids available through Addgene. These plasmids are sufficient to allow construction of WTC_846_ strains carrying the cAR architecture without any further construction. […] Construction of SR strains would require deletion of the negative feedback-controlled TetR from (P2365/2370/2371/2372/2374), and construction of AR strains would require deletion of the constitutively expressed TetR-Tup1 from the same plasmids.”

3. Induction with the WTC_846_ system using aTc is a relatively slow process, at least judged by the Citrine read-out (Figure 4B), taking place over >360 minutes (>3 cell divisions in rich medium). The authors should discuss why this is so, with reference to known transcriptional and/or translational kinetics in this complex feedback system.

We thank the reviewer for directing us to address induction or induction kinetics. We had not done so previously, partly because the story is complicated, and partly because what we knew was probative, not dispositive.

We need to distinguish between speed of induction and time to steady state. Reviewer notes that it takes >3 cell divisions to reach steady state after induction. In WTC_846_ strains, induction, judged by Citrine fluorescence signal (Figure 4B) is detected within 30 minutes after aTc addition, and the release after Cdc20 depletion (Figure 5B) occurs within 35 minutes after aTc addition. Given that induction is rather fast, why do WTC_846_ strains require 6 hours to reach steady state? Expression of the controlled gene is synchronized to expression of the autorepressing TetR. And yet, we know that the long time to steady state is not a consequence of the autorepressing feedback, as we observe that Simple Repression (SR) configurations also require a similar length of time (6 hours) to reach steady state (See Supplementary Figure 6). This time to steady-state derepressed expression is also comparable to that for a previously described TetR based system, which depends on galactose for activation, which required around 400 minutes to reach steady state (doi: 10.1038/nature01546). This time to steady state is still short compared to activation-based systems like the β-estradiol dependent system previously constructed in our lab (doi: 10.1093/nar/gku616).

Our explanation for the long time to equilibrium is that steady state concentration of the controlled protein (Citrine) is reached when degradation of existing protein and its dilution by partition into daughter cells, balances new synthesis. Therefore, the time to steady state will depend on the stability of the controlled protein. If the half-life is longer than doubling time, steady state will be reached in 6-7 hours. If the protein has a half-life shorter than doubling time, steady state will be reached faster. Citrine is long lived and is lost mainly due to dilution by cell division (the longevity of Citrine is shown in our Supplementary Figure S25). Estimates vary as to the stability of the yeast proteome. A recent meta-analysis of 4 different datasets measuring half-lives suggests most proteins (between 50% and 85%) are just as stable and are lost mainly due to dilution by cell division (http://dx.doi.org/10.2139/ssrn.3155916).

Equation 1 is a simple ODE model where a protein is produced at constant rate a and lost with a lumped linear rate (dilution + degradation) d. The analytical solution (Equation 2) of this model shows that the only constant that affects the time variable is the degradation + dilution rate. Dependence on this variable is also evident in the simulations we provide in Appendix 4—figure 1, based on this model. The smaller d is, (i.e., the more stable the protein is), the longer it takes to reach steady state. On the other hand, changes in the production rate a have no effect on time to steady state, although both rates affect the maximum level as one would expect.

In order to make the distinction between induction and steady state kinetics clearer, we made the following changes in the first paragraph of the 4^th^ subsection of results:

Old text read:

“After induction, signal appeared within 30 minutes and reached steady state within 7 hours.”

New text reads:

“After induction, signal appeared within 30 minutes. Time to reach steady state, which will be shorter for proteins that degrade more quickly (see Appendix 4), was 7 hours for the stable protein Citrine.”

We also added the simulations mentioned above as Appendix 4—figure 1, and we refer to it as seen below:

“Time to steady state depends on the stability of the controlled protein. In the experiments in Figure 4, WTC_846_-controlled Citrine takes around 7 hours to reach steady state concentration. […] Therefore most WTC_846_-controlled endogenous proteins will likely have a time to steady state around 7 hours, except those with a shorter half-life which will exhibit a shorter time to steady state.

d[P]dt=a−d•P 𝑃 (1)

[P]= ad(1− e−dt) (2)”

We also made the following addition to the 7^th^ paragraph of the Discussion:

“Both induction and shutoff with WTC_846_ are rapid, as both Citrine and

Cdc20 expression occur within 30 minutes of induction, and shutoff of Citrine expression is observed within 60 minutes. […] Those proteins with shorter halflives will reach steady state faster.”

4. They should also discuss the potential implications of this for screens based upon plate measurements of single-colony growth rates or growth of robotically "pinned" cell cultures. They show a limited number of growth ("spot") assays on solid media that would appear to indicate abrupt cut-offs for colony formation, though the data show only one time point in the growth assay and might obscure growth rate differences.

First, viability experiments. Our plate-based assays using WTC_846_, were not meant to measure growth rates but only the ability of single cells to divide and form colonies (a measure of cell viability) at a terminal time point.

The revised results text in the 3^rd^ paragraph of the last Results section now reads:

“We spotted serial dilutions of cultures of the final seven strains on YPD, YPE, SD, S Glycerol and SD Proline plates, with and without inducer, and assessed the strains' ability to grow into visible colonies at a single time point, at which cells of the parent strain formed colonies in all serially diluted spots(24h for YPD and SD, 42h for others.).”

But this is not what the reviewer asks. We would like to see WTC_846_ used in plate based and robotically pinned growth dependent screens. We know there are differences in time to approach steady state expression for different strains expressing different proteins from WTC_846_ – or by any other inducible system. These differences will lead to different effects on growth rate for the first doublings on solid medium or liquid culture. This fact admits to an obvious workaround, which is to suggest that experimenters pre-induce the strains before plating cells or pinning them robotically into wells on a dish.

In response, we added wording to the last paragraph of the Discussion section about the suitability of WTC_846_ for pinned growth rate experiments. The text now reads:

“Taken together, our results show that WTC_846_ controlled genes define a new type of conditional allele, one that allows precise control of gene dosage.[…] WTC_846_- alleles may find use in engineering applications, for example in directed evolution of higher affinity antibodies expressed in yeast surface display, where precise ability to lower surface concentration should aid selection of….”

5. The authors should examine shut-off kinetics, ideally by as direct a measure of RNA Pol II initiation as possible. This could be of interest in exploring the function of essential genes, for example.

Yes. The reason we were reluctant to explore shutoff kinetics is that it is another complex story. In WTC_846_, elimination of protein from the cell (which is what we care about for assessment of phenotype) depends both on shutoff of new expression and on perdurance.

We expected (and expect) that the speed of shutoff of WTC_846_ controlled genes would be affected primarily by the rates of 3 reversible processes: aTc diffusion out of the cell, TetR binding to its operators, and sequestration of free aTc by newly synthesized TetR. These processes are fast: TetR binding and unbinding to DNA is fast (all binding sites were occupied within ~5 minutes of addition of doxycycline to a reverse TetR system in mammalian cells) (doi: 10.1038/ncomms8357 (2015)). Moreover, in the same experiment, reverse TetR started associating with specific sites seconds after addition of doxycycline, indicating that, at least in mammalian cells inducer diffusion and inducer-TetR binding are also rapid. For this reason (although it is possible that diffusion out of yeast cells might take slightly longer due to the cell wall), we expect that shutoff of WTC_846_-controlled phenotypes will be limited by perdurance, governed by the degradation kinetics of the mRNA and the encoded protein. Given the rapidity of induction (again, fluorescence signal after 30 minutes, Figure 4B, release from CDC20 arrest within 35 minutes, Figure 5E), we reasoned that the same reversible phenomena that govern WTC_846_ induction (namely aTc diffusion, TetR-DNA interactions, and sequestration by newly synthesized TetR) will govern shutoff, so that shutoff of the WTC_846_ expressed phenotype will be dominated by perdurance of the controlled gene products. The consequence would be that for controlled genes (as in Figure 5E with CDC20), the predominant factor in achieving zero protein and observing the associated phenotype in the cell upon aTc removal will be the stability of the mRNA and protein of interest.

Given the relative complexity of this explanation, in response to this reviewer we performed additional direct experiments to (a) determine how quickly WTC_846_ stopped producing the protein of interest upon aTc removal and (b) confirm that perdurance is the predominant factor in how quickly the desired loss of function is seen.

For (a) we grew a strain in which Citrine expression was under WTC_846_ control to exponential phase and then induced with a high concentration (600ng/mL) of aTc. We then measured fluorescence signal every 30 minutes in flow cytometry. Additionally, after 30, 90, 150, and 210 minutes, we removed, washed, and resuspended a sample from the culture in (a) medium without aTc (to shut off WTC_846_) and (b) without aTc but with cyclohexamide (to shut off both WTC_846_ and new protein synthesis). After shutoff, we expected to see an initial increase in signal, followed by decline from this peak. The initial increase in fluorescence after shutoff would be due to 3 factors: the time it took for WTC_846_ to stop producing new transcripts, the time it took for the existing mRNA to be degraded, and continued fluorophore formation by already-synthesized but immature Citrine proteins. Whatever increase in fluorescence we observed above that found in the cycloheximide sample would be due to the other 2 factors outlined above, namely WTC_846_ shutoff speed and mRNA degradation speed.

Our results are presented as a supplementary figure in the revised submission. Shutoff of WTC_846_ directed expression is rapid, as we see stabilization of fluorescence within 60 minutes. And assuming a ~30 minute maturation time for Citrine (since its parent EYFP matures in ~30 minutes after denaturation (https://doi.org/10.1038/nbt0102-87)), WTC_846_ shutoff likely occurs within the first 30 minutes. These results also show that a stable protein like Citrine takes a long time to be eliminated from the cells, given that its level is halved every 90-120 minutes, consistent with dilution through cell division.

In (b), we revisited cell cycle arrest and release by Cdc20. In the original experiment in Figure 5E, arrest of the WTC_846_::CDC20 strain took 8 hours (See Materials and Methods). Given that CDC20 is a very unstable protein, we attributed this long time to arrest to the high aTc concentration in which we precultured the cells (20ng/mL), which likely brought about high pre-arrest concentrations of Cdc20 that took a long time to become depleted. To confirm this hypothesis, we precultured the WTC_846_::CDC20 strain at a much lower aTc concentration (3ng/mL) and then placed cells in medium without aTc. At this lower concentration, arrest began at 1.5 hours after removal of aTc, consistent with the shutoff time of WTC_846_ established above, and between 3-4 hours all cells had arrested. The large difference between the arrest times of cells grown with 3 and 20ng/mL aTc, as well as the short shutoff time of WTC_846_ [WTC_846 –_ Citrine expression] presented above, support the notion that perdurance of protein of interest will in most cases dominate the time to complete “phenotypic” shutoff, i.e. the time needed to see the zero expression "null" phenotype.

We’ve included both experiments as supplementary figures (also presented here) and added the following text to the manuscript.

As the second paragraph of the Results section titled “WTC_846_ fulfils the criteria of an ideal transcriptional controller” we added:

“To better characterize the system, we also measured the shutoff speed of WTC_846_ driven Citrine expression. […] Overall, we conclude that WTC_846_ shutoff is rapid, but the time required to see the phenotypic effects of the absence of the controlled gene product will primarily depend on the stability of the mRNA and expressed protein.”

And we made the following changes in the last paragraph of the Results section titled “WTC_846_ allows precise control over protein dosage and cellular physiology”.

“Finally, we tested the ability of WTC_846_ to exert dynamic control of gene expression by constructing a WTC_846_-K3::CDC20 strain (Y2837) and using this allele to synchronize cells in batch culture by setting Cdc20 expression to zero and then restoring it. […] Given this, and the rapid shutoff kinetics of WTC_846_ presented in Figure 4—figure supplement 8, we conclude that the shutoff dynamics of WTC_846_ controlled phenotypes depend mostly on the speed of degradation of the controlled protein.”

We also added a short comment on the speed of shutoff within the third to last paragraph of the Discussion section:

“Both induction and shutoff with WTC_846_ are rapid, as both Citrine and Cdc20 expression occur within 30 minutes of induction, and shutoff of Citrine expression is observed within 60 minutes.”

We also adjusted our Materials and methods to explain these additional experiments.

6. The expected results for the Whi5 experiment and how they might differ from the actual findings (Figure 5D, pg. 10) are not clear and should be developed further, perhaps in the Results section itself. As the authors point out: "Whi5 controls cell volume by a complex mechanism", but they don't explain what I believe is a key feature, namely that its translation does not scale with cell volume (according to the Skotheim lab), which is an unusual feature. It would be interesting to examine WTC_846_ regulation of other genes implicated in cell size control.

We thank the reviewer for the suggested clarification. We agree that the decoupling of the amount of translation from cell volume is a key feature of Whi5 control. We changed the results (6^th^ paragraph of the last Results section). Revised text now reads.

“Whi5 controls cell volume by a complex mechanism and unlike most other proteins its abundance does not scale with cell volume. […] To test whether we could control cell volume by controlling Whi5, we constructed haploid and diploid WTC_846_::whi5 strains (Y2791, Y2929)…”

In terms of our results, we believe that while we alter the cell cycle stage at which Whi5 mRNA is expressed (by making it expressed continually), the nuclear import/export regulation of Whi5 remains intact. The continued import/ export regulation allows cell size regulation by Whi5 to remain functional, and so that cell volume simply scales with the amount of Whi5 expression we induce via WTC_846._ We clarified this point by adding the text below to the Results:

“Here, we constructed haploid and diploid WTC_846_::WHI5 strains (Y2791, Y2929). […] We expected that the volume of these cells should scale with the concentration of the aTc inducer.”

While we observe that the cell size scaled with aTc concentration, we also found that overexpression of Whi5 in the haploid strain led to increased cell-to-cell variation in cell volume. In order to develop this aspect of the results further, we analysed DNA content of haploid cells grown in presence of high aTc and noticed an increase in the number of cells that had >2n DNA content. We believe this indicates a misregulation of the normal progression of the cell cycle, leading to endoreplication, which could then lead to the observed increase in cell-to-cell variation. We show this experiment in Figure 5—figure supplement 5. We also made the following addition to the Whi5 paragraph of the Results (6^th^ paragraph of the subsection titled “WTC_846_ alleles allow precise control over protein dosage and cellular physiology”):

“Both diploid and haploid cells (especially haploids) expressing high levels of Whi5 showed increased variation in volume. […] We therefore believe that overexpression of Whi5 leads to endoreplication, and the increased variation in volume at high aTc concentrations in the haploid strain originates from these endoreplicated cells.”

It will be interesting to examine WTC_846_ regulation of other cell size control genes, and we are attempting to recruit a student to address this work.

New figure legend reads:

“(C) DNA content of the haploid WTC_846-K1_::WHI5 strain cells (Y2791, red) at mid-exponential phase in S Ethanol media, grown with 4 different aTc concentrations as marked on top of each plot. […] Without Whi5 there was a reduction of G1 cells with 1n DNA content in the Y2791 strain, and upon overexpression of Whi5 aneuploid cells with >2n DNA content are observed.”

7. In Figure 5E, it would be great to show, or at least explain clearly in the text, how the WTC_846_::CDC20 strain compares to the standard cdc20-ts allele used in the field.

We did so. We added the following text together with the relevant references to our Results section to compare our Cdc20 allele to already existing ones:

“When compared to published data, arrest at G2/M using the WTC_846_::CDC20 strain is more penetrant than that obtained using temperature sensitive (~25% unbudded cells) and transcriptionally controlled (~10% unbudded) alleles of cdc20. Release is at least just as fast as that observed for the temperature sensitive (~35 minutes) and the transcriptionally controlled allele (~40 minutes).”

8. In Discussion (pg. 10, bottom right) the authors need to show clearly that WTC_846_ actually gives much lower cell-to-cell variation than all other current methods (at least where this has been measured).

We've done the best we can. When evaluating gene expression tools, cell-to- cell variation has not been routinely measured. This is possibly because all systems thus far have relied on a simple repression or activation configuration, and cell-to-cell variation has been quite high. Moreover, the best means to quantify the different contributions to cell-to-cell variation in expression require microscopic quantification of expression of two reporters in sets of hundreds of single cells tracked over time (Colman-Lerner et al. 2005) and even the best flow cytometric methods for achieving variation (Pesce 2018) involve dual reporter strains rather than the single reporter strains used here.

To address this point, we have included (in references 13, 19, 20), relevant instances in which variation in single reporter strains can be assessed. In these, cell-to-cell variation was not explicitly quantified, but induction resulted in highly variable (even bimodal) distributions of expression. One of those references is to a transcriptional controller induced by β-estradiol (LexA-hER-B112) we published previously. In response to the reviewer's direction, we quantified cell-to-cell variation in gene expression by this controller compared to expression by WTC_846_. The β-estradiol induced system had high cell to cell variation (>0.4) throughout its entire dynamic range, whereas WTC_846_ shows high variation only at a small portion of its dynamic range at low expression levels. Additionally, we constructed a strain where the commonly used, galactose inducible P_GAL1_ promoter directed Citrine expression. We quantified cell-tocell variation in a dose response to galactose, where upon induction variability was always very high (>0.6). We included these comparisons as a supplementary figure.

We also included the following text in our Results section (last paragraph of the section titled “WTC_846_ fulfills the criteria of an ideal transcriptional controller”):

“We quantified the cell-to-cell variability in Citrine expression using the single reporter (ViV) measure for the WTC_846_::citrine strain (Y2759) grown in YPD, and compared it to variation in a β-estradiol (LexA-hER-B112) activation based transcriptional control system we previously described, and the commonly used galactose activated P_GAL1_ promoter(Figure 4D, Figure 4—figure supplements 4, 6 and 9).”

9. In the Discussion (pg. 11 top right) the authors state that "WTC_846_ alleles display no uninduced basal expression" (and earlier in Results they repeated speak of "abolished" basal expression). I think that it would be more correct to point out, at least in theDiscussion, that WTC_846_ alleles display no uninduced basal expression as measured here, by the Citrine fluorescent read-out. Producing null phenotypes in cases where this was not possible with previous systems is a valuable demonstration of the utility of the new method, but not a demonstration of complete repression. The authors might try to measure mRNA steady-state levels by RT-qPCR to estimate copy number per cell, and together with any information on mRNA stability, to estimate transcription rate, if they really want to pin a number on the basal expression level.

The reviewer's qualification is of course correct. Since WTC_846_ is intended as a tool for controlling gene expression, and our functional tests with low expressed genes did not show any problematic basal expression, we opted not to further quantify mRNA, but to change the wording so that it says what we did observe. Discussion now reads.

“It can set protein levels across a large input and output dynamic range. […] WTC_846_ alleles also exhibit high maximum expression, low cell-to-cell variation, and operation in different media conditions without adverse effects on cell physiology.”

Instead of the old text which read:

“It can set protein levels across a large input and output dynamic range. WTC_846_ alleles display no uninduced basal expression, high maximum expression, low cell-to-cell variation, and operate in different media conditions without adverse effects on cell physiology.”

10. Since one of the major potential applications of the WTC_846_ system described here, as pointed out by the authors, is genome-wide screening, that authors should at least discuss how this might be efficiently implemented. The generation of a library of WTC_846_ fusions to each protein-coding gene in yeast by standard methods would be extremely laborious and expensive. An alternative worth mentioning is the SWAp-Tag strategy described by Schuldiner and colleagues (Yofe et al. [2016] Nature Methods). There may very well be others.

We thank the reviewer for the suggested addition. As the reviewer suggested, one good way to generate a library of WTC_846_ controlled endogenous genes would be to use existing whole genome collections such the SWAP-Tag as a starting point. SWAP-Tag allows insertion of N or C-terminal sequences to most ORFs in the yeast genome by creating standard sites for homologous recombination. Generating a whole genome collection of WTC_846_ alleles would be accomplished crossing the existing SWAP-Tag N terminal collection with a WTC_846_ donor strain, followed by sporulation on the appropriate selective medium. We’ve therefore added a short section and the necessary references in our discussion on this subject using the text below.

Last paragraph of the Discussion previously read:

“Taken together, our results show that WTC_846_- controlled genes define a new type of conditional allele that allows precise control of gene dosage. […] We also expect them to find use in genome-wide gene-by-gene and gene-bychemical epistasis screens to detect protein dosage independent interactions,…”

New version reads:

“Taken together, our results show that WTC_846_- controlled genes define a new type of conditional allele, one that allows precise control of gene dosage. *S. cerevisiae* […] We therefore suggest that growth rate-based assays using WTC_846_ or any other inducible system pre-induce cells several generations before plating or pinning. WTC_846_- alleles may find use in engineering applications, for example in directed evolution of higher affinity antibodies expressed in yeast surface display, where precise ability to lower surface concentration should aid selection of….”

11. Although the authors emphasize the utility of WTC_846_ regulation for generating precisely graded expression levels of a given protein with minimal secondary effects growth or physiology, they devoted considerable effort towards achieving maximal shut-off without much discussion of the utility of this feature. In light of anchoring and induced degradation methods that operate at the protein level to generate loss-of-function effects, the authors should consider how WTC_846_ regulation might be a useful complement to these methods, particularly when their efficiency or rapidity is in question.

We thank the reviewer for the suggestion. WTC_846_ could certainly a useful complement to such methods, especially in cases where rapid shutoff of expression is required together with precise dosage control. We have added this to the last part of our discussion using the text below:

“WTC_846_ can also be a useful complement boost the efficiency of methods that act at the protein level such as auxin induced degradation or AnchorAway techniques. […] For example, an experimenter might simultaneously induce depletion of the product of a controlled gene by such a method while adjusting aTc downward to rapidly reset the level of an expressed protein to a new, lower level.”

Additional comments and suggestions1. Pg. 5 right column: "…and we developed a second measure (explained in SI) that normalized variation in dosage with respect to a key confounding variable, cell volume, to correct for its effect on protein concentration. In this, we measured the Residual Standard Deviation (RSD) in signal from cells after normalization of output for volume estimated by a vector of forward and side scatter signals…"I think it would be a good idea to give the second measure a name here and make it clearer that Figure 2D reports the RSD of this value if I understand correctly. The sentence "In this, we measured the RSD in signal…" is awkward and unclear. Please re-word.

Yes. We named the method "Volume Independent Variation". We reworded the unclear sentence as shown below, and clarified in the legend that Figure 2D reports this measure. We modified all other figure legends in the manuscript and the supplementary material where RSD was reported to reflect the new name for the measure.

Old text read:

“In this, we measured RSD in signal from cells after normalization of output for volume estimated by a vector of forward and side scatter signals.”

New text reads:

“In ViV, we estimated cell volume by a vector of forward and side scatter signals, and measured the remaining (Residual) Standard Deviation of the single reporter output after normalization with this estimated volume.”

The new Figure 2 legend now reads:

“(D) Cell-to-cell variation of expression by these three architectures. We calculated single-reporter cell to cell variation (VIV) as described. […] Dot-dash line indicates VIV of the strain where Citrine is constitutively expressed from P_TDH3_ and dashed line indicates VIV of autofluorescence in the parent strain without Citrine.”

2. In Figure 3A the fluorescence curves should be better aligned with the strain designations.

Yes, with thanks. We have made the necessary change to the alignment.

3. Why is there no curve shown for repressed P3tet in Figure 1B? (numbers should be aligned correctly also).

We thank the reviewer for pointing out the alignment issue and have fixed it in the figure.

There is no curve for P3tet because the curve is centred around fluorescence value ~20000 and thus lies outside the range of the xaxis we used in main Figure 1. If we extend the x-axis to include this curve, the difference between all the other curves becomes difficult to perceive. Therefore, we show the data for P3tet separately again in Supplementary Figure 1 (and we direct the reader to the Supplementary Figure at the relevant point in the text).

We have now added a note also in the figure legend (see below) directing the reader to Figure S1.

“Repressed activity of P_3tet_ is above the x axis depicted in this figure, but can be seen in Figure S1.”

4. Yeast gene names should be in italics throughout (including the figure legends). Deletions that a recessive should also by italics, and with the "Δ", essentially an allele designation, placed after the gene name.

Necessary changes made in text and the supplementary material.

5. Reference no. 34 refers to Rap1 and Grf2 binding sites (not Gcr1). Please clarify.

We thank the reviewer for pointing this out. Yes, the paper refers to Rap1 and Grf2 binding sites and finds them in the P_TDH3_ UAS, but also includes in the last portion of its Results section the identification of the CATCC consensus binding sites. The authors of the paper suggest in their discussion that these are likely to be binding sites for the recently discovered Gcr1. This paper was the earliest reference we could find to the CATCC binding sites in the P_TDH3_ UAS, and therefore we cited it in this context.

We added a more recent reference corroborating the GCR1 binding sites in the P _TDH3_ UAS in the main text, next to the original reference 34. We also added a short note “Gcr1 binding sites were found in reference 13 and confirmed in reference 14“ in the legend of Figure S21.

6. Bottom pg. 11, right, I would suggest "…when inducer is absent, the measured basal expression of the controlled gene is abolished…"

We thank the reviewer and have made the suggested change.

7. References no. 28-30 and 35 are incomplete (no journal, volume, or page number).

We thank the reviewer for pointing this out. We fixed the references.

8. Bottom pg. 5, left should be >50,000.

We thank the reviewer for pointing this out. We fixed the symbol.

9. Page 8 right: "…according to,53 to enable…" could be replaced by "…to enable…" with the reference placed at the end of the sentence.

We thank the reviewer for the suggested change which we have implemented in the text.

10. Page 8, right: again, replace inverted "?" with ">" (?)

We thank the reviewer for catching these formatting errors. We have replaced the symbol.

Reviewer #1 (Significance (Required)):The increasing interest in quantitative analysis of cell function emphasizes the need for systems that allow the graded and precise control of gene expression to address dosage effects on phenotype. Although systems of this sort have been developed in yeast, they all have specific drawbacks, as clearly described here. The WTC_846_ system described here represents a genuine improvement on current methods and is likely to be an important addition to the yeast toolbox for gene function analysis. It may also motivate similar developments in mammalian cell culture systems, and eventually in whole animals.

We thank the reviewer for this assessment. Just as qualitative means to bring about conditional gene expression enabled experiments that produced important qualitative insights into cell function, so development of precise quantitative means to bring about conditional gene expression will enable better quantitative analysis of cell function. It will also enable new technical approaches such as the threshold phenotypes, epistasis approaches, and evolutionary methods mentioned above. We suspect that development of analogous methods in in vertebrate cells will likely prove even more important in making vertebrate models of human diseases.

Reviewer #2 (Evidence, reproducibility and clarity (Required)):Summary:Provide a short summary of the findings and key conclusions (including methodology and model system(s) where appropriate).In this manuscript, Azizoglu and co-authors developed an autorepression-based transcriptional controller of gene expression for a given gene of interest in *Saccharomyces cerevisiae*. This system is very precise and allows for dynamic gene expression by adjusting the amount of inducer (aTc) and/or by changing the translation initiation sequences. In addition, it is equally good at being turned on and off in all the media tested and has low cellto-cell variation. The authors tested their transcriptional controller using different genes (including CDC20, IPL1, WHI5, TOR2, and others) and recapitulated various known phenotypes. They also identified a novel phenotype when overexpressing IPL1 in yeast which mimics a previously identified phenotype in mammalian cells when overexpressing Aurora B. This system will be very useful for the yeast community.

We also want to stress that now that we have proof of concept, we can do this in higher cells and multicellular organisms including vertebrates, which need it more.

Major commentsAre the key conclusions convincing?

Yes.

Should the authors qualify some of their claims as preliminary or speculative, or remove them altogether?

No. They do a good job backing up their claims in their supplementary files.

Would additional experiments be essential to support the claims of the paper? Request additional experiments only where necessary for the paper as it is, and do not ask authors to open new lines of experimentation.

No. See above.

Are the suggested experiments realistic in terms of time and resources? It would help if you could add an estimated cost and time investment for substantial experiments.

No suggested experiments.

Are the data and the methods presented in such a way that they can be reproduced?

Yes.

Are the experiments adequately replicated and statistical analysis adequate?

Yes.

Minor comments:Specific experimental issues that are easily addressable. Could the authors add the pGAL1-10-IPL1 control to their WTC_846_-driven IPL1 overexpression experiment (Figure S17)? It might be easier for the reader to understand why the exciting phenotype the authors found was not reported before.

We thank the reviewer for the suggested control. Since we believe that the WTC_846_::IPL1 allele manifests an overexpression phenotype because it produces more protein than P_GAL1_ would have, replacing the promoter of Ipl1 with P_GAL1_ would be the best control. However, since IPL1 is an essential gene, and its misregulation affects chromosomal integrity, we weren't confident we could replace its promoter with P_GAL1_ and still achieve the right expression level to keep the cells healthy. In fact, in previously published work on IPL1, overexpression of IPL1 was achieved by introducing a P_GAL1_-IPL1 producing plasmid into an IPL1+ background.

To address the reviewer's concern we therefore took the same approach. We placed the P_GAL1_ promoter upstream of Citrine in a centromeric plasmid, then grew the strain that carried the (P_GAL1_Citrine) plasmid in the same (Gal Raf) media used by M. MunozBarrera and F. Monje-Casas 2014, to observe effects of Ipl1 overexpression on growth rate and colony formation. The authors did not observe detrimental effects of Ipl1 overexpression in either case.

Here, we used flow cytometry to measure single cell fluorescence from the strain bearing the P_GAL1_-Citrine plasmid, as well as the WTC_846_::citrine strain grown in YPD with 400ng/mL aTc (the same concentration we used in the Ipl1 overexpression experiment).

Our results showed that the median expression level of the centromeric P_GAL1_-Citrine strain was approximately two fold lower than that in the integrated WTC_846_::citrine strain, and that it displayed much higher cell to cell variation. As measured by the single reporter cell-to-cell variation measure (ViV), P_GAL1_-Citrine variation was 0.9, whereas WTC_846_::citrine variation was only 0.19. Ipl1 overexpression phenotype does not lead to a complete arrest of growth in the entire population- some cells are affected while others are not. With PGal1 driven overexpression, overall Ipl1 level is lower compared to WTC_846_ driven overexpression. Additionally, due to high cell-to-cell variability only a handful of cells are actually expressing Ipl1 at this level, which would have made it difficult to observe the growth arrest/delay phenotype that is already not a uniform phenotype in the population. Therefore, we believe the combination of lower expression and higher cell-to-cell variation in the P_GAL1_-Citrine strain likely accounts for the failure of Munoz-Barrera and Monje-Casas to observe a growth arrest phenotype.

We present the results in Figure 5—figure supplement 7. We explain the results in additional sentences at the end of the 4^th^ paragraph of the Results section titled “WTC_846_ alleles allow precise control over protein dosage and cellular physiology“.

The paragraph in the main text now reads:

“At high aTc concentrations, the WTC_846_-K2:IPL1 strain formed colonies with lower plating efficiency than the parent strain. […] Either the lower expression or the higher variation, or both, might account for the fact that P_GAL1_ driven Ipl1 overexpression does not result in the mammalian Aurora B phenotype in *S. cerevisiae*."

Are prior studies referenced appropriately?

Yes.

Are the text and figures clear and accurate? The authors used the strain names in the text and it is hard to figure out what each strain contains. They could transmit this information by using the construction "strain Y### (contains [specify relevant details]).

We thank the reviewer for the suggestion. During the revision process, we took care to mention the relevant details as much as possible together with the yeast strain number within the text.

In Figure 3, specify that the sentence that explains the errors bars refers to section A and B. or move the sentence before C.

Right. There are actually error bars also around the data points in C, although they are very narrow and difficult to see. This sentence therefore refers to Figure 3C as well. In response to this comment, we adapted the figure to make it easier to see the error bars in Figure 3C.

In page 3, the authors mention ts and cs mutations. The authors should define what ts and cs are and italicize them.

We thank the reviewer for pointing this out. We have included the long format of these abbreviations where they were first mentioned, and have italicized the abbreviations.

Old text read:

“During the 20th century, workhorse methods to ensure the presence or absence of gene products have included use of ts and cs mutations within genes, for example to give insight into ordinality of cell biological events.”

New text reads:

“During the 20th century, workhorse methods to ensure the presence or absence of gene products have included use of temperature sensitive (ts) and cold sensitive (cs) mutations within genes, for example to give insight into ordinality of cell biological events.”

On page 4, the authors defined PIC and aTc, but both terms were previously defined.

Fixed.

On page 4, in the following sentence "We also combined the operators in theseconstructs to generate P5tet (Figure 2B)", the figure is wrongly referenced, the reference should be to Figure 1B.

We thank the reviewer for catching this, and have fixed the reference.

On page 7, in the following sentence "The PACT1, PVPH1, PRNR2 strains showed no uninduced expression,.…" please add a reference to Figure 3B.

We thank the reviewer for the suggestion and have added the reference.

On page 10, in the following sentence "…at high aTc concentrations, cells bearing the higher translational efficiency TOR2 allele (WTC_846_-K1::TOR2) grew more slowly than the otherwise-isogenic control parent strain with WT TOR2 (Figure S18C)" the figure is wrongly referenced, the reference should be to Figure S18D. I also suggest the authors add the following text: (Figure S18D, compare the 600ng/mL line to blue dashed line).

We thank the reviewer for pointing this out and have corrected the reference. We also added the suggested text for clarification.

On page 10, in the following sentence "…Whi5 is rapidly exported from the nucleus, and the cells enter START", I think they mean S-phase.

We thank the author for the comment and have made the correction to the text which now says S-phase.

On page 10, the Kozak sequence is missing in the following sentence: "Use of WTC_846_::CDC20 to synchronize the cells…"

We thank the reviewer. We have added the correct Kozak sequence information to the strain name.

Throughout the paper, gene deletions should be written as follow: whi5Δ.Throughout the paper, genes should be italicized (e.g. HIS3).

We thank the reviewer for the correction of the naming nomenclature. We have fixed the gene names throughout the paper and the supplementary material.

Throughout the paper, the authors have random question marks (which may be the result of converting a file in another format to a PDF) that may be removed.

We thank the reviewer for catching these formatting errors. We replaced the question marks with the correct symbols.

On Supplementary Text page 1, the following sentence: "However single reporter studies have a major, confounding contribution to measured variation in gene expression that multi-reporter studies don't:" is missing a comma after However and change don't to do not.

We thank the reviewer for both corrections which we have implemented as suggested.

In Supplemental Figure 7, the authors may want to add P2tet and P3tet for better main text that the promoter for essential genes was replaced with another promoter in the presence of the inducer (aTc) before referencing Figure 5, or in the legend of figure 5.

We believe the reviewer wanted to raise two separate points here. One about Figure S7, the other about clarifying the essential endogenous gene promoter replacement protocol we used. We address these two points separately below.

First. We thank the reviewer for the suggestion about Figure S7. In Figure S7, inclusion of the unrepressed promoter fluorescence curves would widen the x-axis range we need to show, and therefore would make it more difficult for the reader to perceive the small differences between the repressed expression of P_2tet_ and P_3tet_ compared to the autofluorescence. Therefore, given that the focus of this supplementary figure is narrowly on the repressed expression, we believe it is best to omit the unrepressed expression curves in this figure.

It is important to note that, although P_3tet_ is weaker than P_2tet_, this fact is not the only reason that P_3tet_ shows a lower level of repressed expression. Unrepressed P_2tet_ is 1.14 times stronger than P_3tet_ (data from main figure 1). However, when repressed, P_2tet_ expression is 1.38 higher than that for P_3tet_. This suggests that promoter strength is not the explanation for the lower repressed expression from P_3tet_. To emphasize this point, in response to the reviewer direction, we added fold comparison numbers to the figure legend.

The new text in the figure legend of Supplementary Figure 7 reads as follows:

“For comparison, repressed expression in the P_2tet_ strain (Y2662) was 1.38 fold higher than in the P_3tet_ strain (Y2702), even though the expression in the unrepresssed P_2tet_ strain was only 1.14 fold more than P_3tet_. This fact shows that, even given the difference in promoter strength, P_3tet_ is repressed more effectively than P_2tet_.”

Second, we thank the reviewer for the suggested clarification to the strain generation protocol. We changed the main text to indicate that the strains were generated by replacing the promoter of the endogenous gene in cells growing in medium containing aTc., using the text presented below:

“We constructed strains in which WTC_846_ controlled the expression of these genes by replacing the promoter of the endogenous gene. Before transformation the cells were grown in liquid medium containing aTc, and then plated on solid medium containing aTc (see Supplementary Materials for a detailed protocol).”

Reviewer #2 (Significance (Required)):Describe the nature and significance of the advance (e.g. conceptual, technical, clinical) for the field. The authors provide a new tool to dynamically control the expression of genes. The previous systems that have been developed are hard to modulate or are "leaky". The system developed in this work allows for precise control of gene dosage, has low cell-to-cell variation, and has a true OFF state.

Static steady state expression around an adjustable setpoint, i.e. clamped expression is good too.

Place the work in the context of the existing literature (provide references, where appropriate).Regulating gene expression in a targeted way has been a common method in yeast research for decades. However, the current gene expression systems have deficiencies (e.g. leaky expression, not tunable expression, high cell-to-cell variation). The authors in this paper developed a new system that shore up these deficiencies by enabling tight on and off controlled gene expression.State what audience might be interested in and influenced by the reported findings. The yeast community in general, as it is a transcriptional controller developed for *S. cerevisiae*.

We also believe the ability to make alleles of this type will be extended higher cells and multicellular organisms, particularly vertebrates.

Define your field of expertise with a few keywords to help the authors contextualize your point of view. Indicate if there are any parts of the paper that you do not have sufficient expertise to evaluate. Genetics, yeast, evolution, chromosome segregation.